# Algorithms and Theory for Multiple-Source Adaptation

**Judy Hoffman**
CS Department UC Berkeley
Berkeley, CA 94720
jhoffman@eecs.berkeley.edu

**Mehryar Mohri**
Courant Institute and Google
New York, NY 10012
mohri@cims.nyu.edu

**Ningshan Zhang**
New York University
New York, NY 10012
nzhang@stern.nyu.edu

## Abstract

We present a number of novel contributions to the multiple-source adaptation problem. We derive new normalized solutions with strong theoretical guarantees for the cross-entropy loss and other similar losses. We also provide new guarantees that hold in the case where the conditional probabilities for the source domains are distinct. Moreover, we give new algorithms for determining the distribution-weighted combination solution for the cross-entropy loss and other losses. We report the results of a series of experiments with real-world datasets. We find that our algorithm outperforms competing approaches by producing a single robust model that performs well on any target mixture distribution. Altogether, our theory, algorithms, and empirical results provide a full solution for the multiple-source adaptation problem with very practical benefits.

## 1 Introduction

In many modern applications, often the learner has access to information about several source domains, including accurate predictors possibly trained and made available by others, but no direct information about a target domain for which one wishes to achieve a good performance. The target domain can typically be viewed as a combination of the source domains, that is a mixture of their joint distributions, or it may be close to such mixtures. In addition, often the learner does not have access to all source data simultaneously, for legitimate reasons such as privacy or storage limitation. Thus, the learner cannot simply pool all source data together to learn a predictor.

Such problems arise commonly in speech recognition where different groups of speakers (domains) yield different acoustic models and the problem is to derive an accurate acoustic model for a broader population that may be viewed as a mixture of the source groups (Liao, 2013). In object recognition, multiple image databases exist, each with its own bias and labeled categories (Torralba and Efros, 2011), but the target application may contain images which most closely resemble only a subset of the available training data. Finally, in sentiment analysis, accurate predictors may be available for sub-domains such as TVs, laptops and CD players, each previously trained on labeled data, but no labeled data or predictor may be at the learner's disposal for the more general category of electronics, which can be modeled as a mixture of the sub-domains (Blitzer et al., 2007; Dredze et al., 2008).

The problem of transfer from a single source to a known target domain (Ben-David, Blitzer, Crammer, and Pereira, 2006; Mansour, Mohri, and Rostamizadeh, 2009b; Cortes and Mohri, 2014; Cortes, Mohri, and Muñoz Medina, 2015), either through unsupervised adaptation techniques (Gong et al., 2012; Long et al., 2015; Ganin and Lempitsky, 2015; Tzeng et al., 2015), or via lightly supervised ones (some amount of labeled data from the target domain) (Saenko et al., 2010; Yang et al., 2007; Hoffman et al., 2013; Girshick et al., 2014), has been extensively investigated in the past. Here, we focus on the problem of multiple-source domain adaptation and ask how the learner can combine relatively accurate predictors available for each source domain to derive an accurate predictor for

*any* new mixture target domain? This is known as the *multiple-source adaptation (MSA) problem* first formalized and analyzed theoretically by Mansour, Mohri, and Rostamizadeh (2008, 2009a) and later studied for various applications such as object recognition (Hoffman et al., 2012; Gong et al., 2013a,b). Recently, Zhang et al. (2015) studied a causal formulation of this problem for a classification scenario, using the same combination rules as Mansour et al. (2008, 2009a). A closely related problem to the MSA problem is that of domain generalization (Pan and Yang, 2010; Muandet et al., 2013; Xu et al., 2014), where knowledge from an arbitrary number of related domains is combined to perform well on a previously unseen domain. Appendix G includes a more detailed discussion of previous work related to the MSA problem.

Mansour, Mohri, and Rostamizadeh (2008, 2009a) gave strong theoretical guarantees for a distribution-weighted combination to address the MSA problem, but they did not provide an algorithmic solution to determine that combination. Furthermore, the solution they proposed could not be used for loss functions such as cross-entropy, which require a normalized predictor. Their work also assumed a deterministic scenario (non-stochastic) with the same labeling function for all source domains.

This work makes a number of novel contributions to the MSA problem. We give new normalized solutions with strong theoretical guarantees for the cross-entropy loss and other similar losses. Our guarantees hold even when the conditional probabilities for the source domains are distinct. A by-product of our analysis is the extension of the theoretical results of Mansour et al. (2008, 2009a) to the stochastic scenario, where there is a joint distribution over the input and output space.

Moreover, we give new algorithms for determining the distribution-weighted combination solution for the cross-entropy loss and other losses. We prove that the problem of determining that solution can be cast as a DC-programming (difference of convex) and prove explicit DC-decompositions for the cross-entropy loss and other losses. We also give a series of experimental results with several datasets demonstrating that our distribution-weighted combination solution is remarkably robust. Our algorithm outperforms competing approaches and performs well on any target mixture distribution.

Altogether, our theory, algorithms, and empirical results provide a full solution for the MSA problem with very practical benefits.

## 2  Problem setup

Let $\mathcal{X}$ denote the input space and $\mathcal{Y}$ the output space. We consider a multiple-source domain adaptation (MSA) problem in the general stochastic scenario where there is a distribution over the joint input-output space $\mathcal{X} \times \mathcal{Y}$. This is a more general setup than the deterministic scenario in (Mansour et al., 2008, 2009a), where a target function mapping from $\mathcal{X}$ to $\mathcal{Y}$ is assumed. This extension is needed for the analysis of the most common and realistic learning setups in practice. We will assume that $\mathcal{X}$ and $\mathcal{Y}$ are discrete, but the predictors we consider can take real values. Our theory can be straightforwardly extended to the continuous case with summations replaced by integrals in the proofs. We will identify a *domain* with a distribution over $\mathcal{X} \times \mathcal{Y}$ and consider the scenario where the learner has access to a predictor $h_k$, for each domain $\mathcal{D}_k$, $k \in [p] = \{1, \ldots, p\}$.

We consider two types of predictor functions $h_k$, and their associated loss functions $L$ under the regression model (R) and the probability model (P) respectively,

$$
\begin{array}{lll}
h_k \colon \mathcal{X} \to \mathbb{R} & L \colon \mathbb{R} \times \mathcal{Y} \to \mathbb{R}_+ & (R) \\
h_k \colon \mathcal{X} \times \mathcal{Y} \to [0,1] & L \colon [0,1] \to \mathbb{R}_+ & (P)
\end{array}
$$

We abuse the notation and write $L(h, x, y)$ to denote the loss of a predictor $h$ at point $(x, y)$, that is $L(h(x), y)$ in the regression model, and $L(h(x, y))$ in the probability model. We will denote by $\mathcal{L}(\mathcal{D}, h)$ the expected loss of a predictor $h$ with respect to the distribution $\mathcal{D}$:

$$
\mathcal{L}(\mathcal{D}, h) = \mathbb{E}_{(x,y) \sim \mathcal{D}} \big[ L(h, x, y) \big] = \sum_{(x,y) \in \mathcal{X} \times \mathcal{Y}} \mathcal{D}(x, y) L(h, x, y).
$$

Much of our theory only assumes that $L$ is convex and continuous. But, we will be particularly interested in the case where, in the regression model, $L(h(x), y) = (h(x) - y)^2$ is the squared loss, and where, in the probability model, $L(h(x, y)) = -\log h(x, y)$ is the cross-entropy loss (log-loss).

We will assume that each $h_k$ is a relatively accurate predictor for the distribution $\mathcal{D}_k$: there exists $\epsilon > 0$ such that $\mathcal{L}(\mathcal{D}_k, h_k) \leq \epsilon$ for all $k \in [p]$. We will also assume that the loss of the source hypotheses $h_k$ is bounded, that is $L(h_k, x, y) \leq M$ for all $(x, y) \in \mathcal{X} \times \mathcal{Y}$ and all $k \in [p]$.

In the MSA problem, the learner's objective is to combine these predictors to design a predictor with small expected loss on a target domain that could be an arbitrary and unknown mixture of the source domains, the case we are particularly interested in, or even some other arbitrary distribution. It is worth emphasizing that the learner has no knowledge of the target domain.

How do we combine the $h_k$s? Can we use a convex combination rule, $\sum_{k=1}^{p} \lambda_k h_k$, for some $\lambda \in \Delta$? In Appendix A (Lemmas 9 and 10) we show that *no* convex combination rule will perform well even in very simple MSA problems. These results generalize a previous lower bound of Mansour et al. (2008). Next, we show that the distribution-weighted combination rule is a suitable solution.

Extending the definition given by Mansour et al. (2008), we define the distribution-weighted combination of the functions $h_k$, $k \in [p]$ as follows. For any $\eta > 0$, $z \in \Delta$, and $(x, y) \in \mathcal{X} \times \mathcal{Y}$,

$$h_z^{\eta}(x) = \sum_{k=1}^{p} \frac{z_k \mathcal{D}_k^1(x) + \eta \frac{\mathcal{U}^1(x)}{p}}{\sum_{k=1}^{p} z_k \mathcal{D}_k^1(x) + \eta \mathcal{U}^1(x)} h_k(x), \qquad (R) \qquad (1)$$

$$h_z^{\eta}(x, y) = \sum_{k=1}^{p} \frac{z_k \mathcal{D}_k(x, y) + \eta \frac{\mathcal{U}(x,y)}{p}}{\sum_{j=1}^{p} z_j \mathcal{D}_j(x, y) + \eta \mathcal{U}(x, y)} h_k(x, y), \qquad (P) \qquad (2)$$

where we denote by $\mathcal{D}^1$ the marginal distribution over $\mathcal{X}$, for all $x \in \mathcal{X}$, $\mathcal{D}^1(x) = \sum_{y \in \mathcal{Y}} \mathcal{D}(x, y)$, and by $\mathcal{U}^1$ the uniform distribution over $\mathcal{X}$. This extension may seem technically straightforward in hindsight, but the form of the predictor was not immediately clear in the stochastic case.

## 3 Theoretical guarantees

In this section, we present a series of theoretical guarantees for distribution-weighted combinations with a suitable choice of the parameters $z$ and $\eta$, both for the regression model and for the probability model. We first give our main result for the general stochastic scenario. Next, for the probability model with cross-entropy loss, we introduce a *normalized* distribution weighted combination and prove that it benefits from strong theoretical guarantees.

Our theoretical results rely on a measure of divergence between two distributions. The one that naturally comes up in our analysis is the *Rényi Divergence* (Rényi, 1961). We will denote by $\mathsf{d}_\alpha(\mathcal{D} \parallel \mathcal{D}') = e^{\mathsf{D}_\alpha(\mathcal{D} \parallel \mathcal{D}')}$ the exponential of the $\alpha$-Rényi Divergence of two distributions $\mathcal{D}$ and $\mathcal{D}'$. See Appendix F for more details about the notion of Rényi Divergence.

### 3.1 General guarantees for regression and probability models

Let $\mathcal{D}_T$ be an unknown target distribution. We will denote by $\mathcal{D}_T(\cdot|x)$ and $\mathcal{D}_k(\cdot|x)$ the conditional probability distribution on the target and the source domain $k$ respectively. We do not assume that the target and source conditional probabilities $\mathcal{D}_T(\cdot|x)$ and $\mathcal{D}_k(\cdot|x)$ coincide for all $k \in [p]$ and $x \in \mathcal{X}$. This is a significant extension of the MSA scenario with respect to the one considered by Mansour et al. (2009a), which assumed exactly the same labeling function $f$ for all source domains, in the deterministic scenario.

Let $\mathcal{D}_T$ be a mixture of source distributions, such that $\mathcal{D}_T^1 \in \mathcal{D}^1 = \{\sum_{k=1}^{p} \lambda_k \mathcal{D}_k^1 : \lambda \in \Delta\}$ in the regression model, or $\mathcal{D}_T \in \mathcal{D} = \{\sum_{k=1}^{p} \lambda_k \mathcal{D}_k : \lambda \in \Delta\}$ in the probability model. We also assume that under the regression model, all possible target distributions $\mathcal{D}_T$ admit the same (unknown) conditional probability distribution.

Fix $\alpha > 1$ and define $\epsilon_T$ by

$$\epsilon_T = \max_{k \in [p]} \left[ \underset{x \sim \mathcal{D}_k^1}{\mathbb{E}} \left[ \mathsf{d}_\alpha \left( \mathcal{D}_T(\cdot|x) \parallel \mathcal{D}_k(\cdot|x) \right)^{\alpha-1} \right] \right]^{\frac{1}{\alpha}} \epsilon^{\frac{\alpha-1}{\alpha}} M^{\frac{1}{\alpha}}.$$

$\epsilon_T$ depends on the maximal expected Rényi divergence between the target conditional probability distribution $\mathcal{D}_T(\cdot|x)$ and the source ones $\mathcal{D}_k(\cdot|x)$, $\forall k \in [p]$, with the expectation taken over the source marginal distribution $\mathcal{D}_k^1$, and the maximum taken over $k \in [p]$. When the target conditional is close to all source ones, $\alpha$ can be chosen to be very large and $\epsilon_T$ is close to $\epsilon$. In particular, when the conditional probabilities coincide, for $\alpha = +\infty$, we have $\epsilon_T = \epsilon$.

**Theorem 1.** *For any $\delta > 0$, there exist $\eta > 0$ and $z \in \Delta$ such that the following inequalities hold for any $\alpha > 1$ and any target distribution $\mathcal{D}_T$ that is a mixture of source distributions:*

$$\mathcal{L}(\mathcal{D}_T, h_z^\eta) \le \epsilon_T + \delta, \qquad\qquad (R)$$
$$\mathcal{L}(\mathcal{D}_T, h_z^\eta) \le \epsilon + \delta. \qquad\qquad (P)$$

As discussed later, the proof of more general results (Theorem 2 and Theorem 14) is given in Appendix B. The learning guarantees for the regression and the probability model are slightly different, since the definitions of the distribution-weighted combinations are different for the two models. Theorem 1 shows the existence of $\eta > 0$ and a mixture weight $z \in \Delta$ with a remarkable property: in the regression model (R), for any target distribution $\mathcal{D}_T$ whose conditional $\mathcal{D}_T(\cdot|x)$ is on average not too far away from $\mathcal{D}_k(\cdot|x)$ for any $k \in [p]$, and $\mathcal{D}_T^1 \in \mathcal{D}^1$, the loss of $h_z^\eta$ on $\mathcal{D}_T$ is small. It is even more remarkable that, in the probability model (P), the loss of $h_z^\eta$ is at most $\epsilon$ on any target distribution $\mathcal{D}_T \in \mathcal{D}$. Thus, $h_z^\eta$ is a robust hypothesis with favorable property for any such target distribution $\mathcal{D}_T$.

We now present a more general result, Theorem 2, that relaxes the assumptions under the regression model that all possible target distributions $\mathcal{D}_T$ admit the same conditional probability distribution, and that the target's marginal distribution is a mixture of source ones. In Appendix B, we show that Theorem 2 coincides with Theorem 1 under those assumptions. In Appendix B, we further give a more general result than Theorem 1 under the probability model (Theorem 14).

To present this more general result, we first introduce some additional notation. Given a conditional probability distribution $\mathcal{Q}(\cdot|x)$ defined for all $x \in \mathcal{X}$, define $\epsilon_\alpha(\mathcal{Q})$ as follows:

$$\epsilon_\alpha(\mathcal{Q}) = \max_{k \in [p]} \left[ \mathbb{E}_{x \sim \mathcal{D}_k^1} \left[ \mathsf{d}_\alpha \left( \mathcal{Q}(\cdot|x) \, \| \, \mathcal{D}_k(\cdot|x) \right)^{\alpha-1} \right] \right]^{\frac{1}{\alpha}} \epsilon^{\frac{\alpha-1}{\alpha}} M^{\frac{1}{\alpha}}.$$

Thus, $\epsilon_\alpha(\mathcal{Q})$ depends on the maximal expected $\alpha$-Rényi divergence between $\mathcal{Q}(\cdot|x)$ and $\mathcal{D}_k(\cdot|x)$, and $\epsilon_\alpha(\mathcal{Q}) = \epsilon_T$ when $\mathcal{Q}(\cdot|x) = \mathcal{D}_T(\cdot|x)$. When there exists $\mathcal{Q}(\cdot|x)$ such that the expected $\alpha$-Rényi divergence is small for all $k \in [p]$, then $\epsilon_\alpha(\mathcal{Q})$ is close to $\epsilon$ for $\alpha = +\infty$. In addition, we will use the following definitions: $\mathcal{D}_{k,\mathcal{Q}}(x,y) = \mathcal{D}_k^1(x)\mathcal{Q}(y|x)$ and $\mathcal{D}_{P,\mathcal{Q}} = \left\{ \sum_{k=1}^p \lambda_k \mathcal{D}_{k,\mathcal{Q}} : \lambda \in \Delta \right\}$.

**Theorem 2** (Regression model). *Fix a conditional probability distribution $\mathcal{Q}(\cdot|x)$ defined for all $x \in \mathcal{X}$. Then, for any $\delta > 0$, there exist $\eta > 0$ and $z \in \Delta$ such that the following inequality holds for any $\alpha, \beta > 1$ and any target distribution $\mathcal{D}_T$:*

$$\mathcal{L}(\mathcal{D}_T, h_z^\eta) \le \left[ \left( \epsilon_\alpha(\mathcal{Q}) + \delta \right) \mathsf{d}_\beta(\mathcal{D}_T \, \| \, \mathcal{D}_{P,\mathcal{Q}}) \right]^{\frac{\beta-1}{\beta}} M^{\frac{1}{\beta}}.$$

The learning guarantee of Theorem 2 depends on the Rényi divergence between the conditional probabilities of the source and target domains and a fixed *pivot* $\mathcal{Q}(\cdot|x)$. In particular, when there exists a pivot $\mathcal{Q}(\cdot|x)$ that is close to $\mathcal{D}_T(\cdot|x)$ and $\mathcal{D}_k(\cdot|x)$, for all $k \in [p]$, then the guarantee is significant. One candidate for such a pivot is a conditional probability distribution $\mathcal{Q}(\cdot|x)$ minimizing $\epsilon_\alpha(\mathcal{Q})$.

In many learning tasks, it is reasonable to assume that the conditional probability of the output labels is the same across source domains. For example, a dog picture represents a dog regardless of whether the picture belongs to an individual's personal collection or to a broader database of pictures from multiple individuals. This is a straightforward extension of the assumption adopted by Mansour et al. (2008) in the deterministic scenario, where exactly the same labeling function $f$ is assumed for all source domains. In that case, we have $\mathcal{D}_T(\cdot|x) = \mathcal{D}_k(\cdot|x)$, $\forall k \in [p]$ and therefore $\mathsf{d}_\alpha(\mathcal{D}_T(\cdot|x) \, \| \, \mathcal{D}_k(\cdot|x)) = 1$. Setting $\alpha = +\infty$, we recover the main result of Mansour et al. (2008).

**Corollary 3.** *Assume that the conditional probability distributions $\mathcal{D}_k(\cdot|x)$ do not depend on $k$. Then, for any $\delta > 0$, there exist $\eta > 0$ and $z \in \Delta$ such that $\mathcal{L}(\mathcal{D}_\lambda, h_z^\eta) \le \epsilon + \delta$ for any mixture parameter $\lambda \in \Delta$.*

Corollary 3 shows the existence of a parameter $\eta > 0$ and a mixture weight $z \in \Delta$ with a remarkable property: for any $\delta > 0$, regardless of which mixture weight $\lambda \in \Delta$ defines the target distribution, the loss of $h_z^\eta$ is at most $\epsilon + \delta$, that is arbitrarily close to $\epsilon$. $h_z^\eta$ is therefore a *robust* hypothesis with a favorable property for any mixture target distribution.

To cover the realistic cases in applications, we further extend this result to the case where the distributions $\mathcal{D}_k$ are not directly available to the learner, and instead estimates $\widehat{\mathcal{D}}_k$ have been derived

from data, and further to the case where the target distribution $\mathcal{D}_T$ is not a mixture of source distributions. We will denote by $\widehat{h}_z^\eta$ the distribution-weighted combination rule based on the estimates $\widehat{\mathcal{D}}_k$. Our learning guarantee for $\widehat{h}_z^\eta$ depends on the Rényi divergence of $\widehat{\mathcal{D}}_k$ and $\mathcal{D}_k$, as well as the Rényi divergence of $\mathcal{D}_T$ and the family of mixtures of source distributions.

**Corollary 4.** *For any $\delta > 0$, there exist $\eta > 0$ and $z \in \Delta$, such that the following inequality holds for any $\alpha > 1$ and arbitrary target distribution $\mathcal{D}_T$:*

$$\mathcal{L}(\mathcal{D}_T, \widehat{h}_z^\eta) \le \left[ (\widehat{\epsilon} + \delta) \, \mathsf{d}_\alpha(\mathcal{D}_T \parallel \widehat{\mathcal{D}}) \right]^{\frac{\alpha-1}{\alpha}} M^{\frac{1}{\alpha}},$$

*where $\widehat{\epsilon} = \max_{k \in [p]} \left[ \epsilon \, \mathsf{d}_\alpha(\widehat{\mathcal{D}}_k \parallel \mathcal{D}_k) \right]^{\frac{\alpha-1}{\alpha}} M^{\frac{1}{\alpha}}$, and $\widehat{\mathcal{D}} = \left\{ \sum_{k=1}^p \lambda_k \widehat{\mathcal{D}}_k \colon \lambda \in \Delta \right\}$.*

Corollary 4 shows that there exists a predictor $\widehat{h}_z^\eta$ based on the estimate distributions $\widehat{\mathcal{D}}_k$ that is $\widehat{\epsilon}$-accurate with respect to any target distribution $\mathcal{D}_T$ whose Rényi divergence with respect to the family $\widehat{\mathcal{D}}$ is not too large ($\mathsf{d}_\alpha(\mathcal{D}_T \parallel \widehat{\mathcal{D}})$ close to 1). Furthermore, $\widehat{\epsilon}$ is close to $\epsilon$, provided that $\widehat{\mathcal{D}}_k$s are good estimates of $\mathcal{D}_k$s (that is $\mathsf{d}_\alpha(\widehat{\mathcal{D}}_k \parallel \mathcal{D}_k)$ close to 1). The proof is given in Appendix B.

### 3.2 Guarantees for the probability model with the cross-entropy loss

Here, we discuss the important special case where $L$ coincides with the cross-entropy loss in the probability model, and present a guarantee for a normalized distribution-weighted combination solution. This analysis is a complement to Theorem 1, which only holds for the unnormalized hypothesis $h_z^\eta(x, y)$.

The cross-entropy loss assumes normalized hypotheses. Thus, here, we assume that the source functions are normalized for every $x$: $\sum_{y \in \mathcal{Y}} h_k(x, y) = 1$, $\forall x \in \mathcal{X}, \forall k \in [p]$. For any $\eta > 0$ and $z \in \Delta$, we define a normalized weighted combination $\overline{h}_z^\eta(x, y)$ that is based on distribution-weighted combination $h_z^\eta(x, y)$ defined by (2):

$$\overline{h}_z^\eta(x, y) = \frac{h_z^\eta(x, y)}{\sum_{y \in \mathcal{Y}} h_z^\eta(x, y)}.$$

We will first assume the conditional probability distributions $\mathcal{D}_k(\cdot | x)$ do not depend on $k$.

**Theorem 5.** *Assume that there exists $\mu > 0$ such that $\mathcal{D}_k(x, y) \ge \mu \mathcal{U}(x, y)$ for all $k \in [p]$ and $(x, y) \in \mathcal{X} \times \mathcal{Y}$. Then, for any $\delta > 0$, there exist $\eta > 0$ and $z \in \Delta$ such that $\mathcal{L}(\mathcal{D}_\lambda, \overline{h}_z^\eta) \le \epsilon + \delta$ for any mixture parameter $\lambda \in \Delta$.*

Theorem 5 provides a strong guarantee that is the analogue of Corollary 3 for normalized distribution-weighted combinations. The theorem can also be extended to the case of arbitrary target distributions and estimated densities. When the conditional probabilities are distinct across the source domains, we propose a marginal distribution-weighted combination rule, which is already normalized. We can directly apply Theorem 1 to that solution and achieve favorable guarantees. More details are presented in Appendix C.

These results are non-trivial and important, as they provide a guarantee for an accurate and robust predictor for a commonly used loss function, the cross-entropy loss.

## 4 Algorithms

We have shown that, for both the regression and the probability model, there exists a vector $z$ defining a distribution-weighted combination hypothesis $h_z^\eta$ that admits very favorable guarantees. But how can we find a such $z$? This is a key question in the MSA problem which was not addressed by Mansour et al. (2008, 2009a): no algorithm was previously reported to determine the mixture parameter $z$, even for the deterministic scenario. Here, we give an algorithm for determining that vector $z$.

In this section, we give practical and efficient algorithms for finding the vector $z$ in the important cases of the squared loss in the regression model, or the cross-entropy loss in the probability model, by leveraging the differentiability of the loss functions. We first show that $z$ is the solution of a general optimization problem. Next, we give a DC-decomposition (difference of convex decomposition)

of the objective for both models, thereby proving an explicit DC-programming formulation of the problem. This leads to an efficient DC algorithm that is guaranteed to converge to a stationary point. Additionally, we show that it is straightforward to test if the solution obtained is the global optimum. While we are not proving that the local stationary point found by our algorithm is the global optimum, empirically, we observe that that is indeed the case.

## 4.1 Optimization problem

Theorem 1 shows that the hypothesis $h_z^\eta$ based on the mixture parameter $z$ benefits from a strong generalization guarantee. A key step in proving Theorem 1 is to show the following lemma.

**Lemma 6.** *For any $\eta, \eta' > 0$, there exists $z \in \Delta$, with $z_k \neq 0$ for all $k \in [p]$, such that the following holds for the distribution-weighted combining rule $h_z^\eta$:*

$$\forall k \in [p], \quad \mathcal{L}(\mathcal{D}_k, h_z^\eta) \leq \sum_{j=1}^{p} z_j \mathcal{L}(\mathcal{D}_j, h_z^\eta) + \eta'. \tag{3}$$

Lemma 6 indicates that for the solution $z$, $h_z^\eta$ has essentially the same loss on all source domains. Thus, our problem consists of finding a parameter $z$ verifying this property. This, in turn, can be formulated as a min-max problem: $\min_{z \in \Delta} \max_{k \in [p]} \mathcal{L}(\mathcal{D}_k, h_z^\eta) - \mathcal{L}(\mathcal{D}_z, h_z^\eta)$, which can be equivalently formulated as the following optimization problem:

$$\min_{z \in \Delta, \gamma \in \mathbb{R}} \gamma \quad \text{s.t. } \mathcal{L}(\mathcal{D}_k, h_z^\eta) - \mathcal{L}(\mathcal{D}_z, h_z^\eta) \leq \gamma, \forall k \in [p]. \tag{4}$$

## 4.2 DC-decomposition

We provide explicit DC decompositions of the objective of Problem (4) for the regression model with the squared loss and for the probability model with the cross-entropy loss. The derivations are given in Appendix D. We first rewrite $h_z^\eta$ as the division of two affine functions for both the regression (R) and the probability (P) model, $h_z = J_z/K_z$, where we adopt the following definitions and notation:

$$J_z(x) = \sum_{k=1}^{p} z_k \mathcal{D}_k^1(x) h_k(x) + \frac{\eta}{p} \mathcal{U}^1(x) h_k(x), \qquad K_z(x) = \mathcal{D}_z^1(x) + \eta \mathcal{U}^1(x), \tag{R}$$

$$J_z(x,y) = \sum_{k=1}^{p} z_k \mathcal{D}_k(x,y) h_k(x,y) + \frac{\eta}{p} \mathcal{U}(x,y) h_k(x,y), \quad K_z(x,y) = D_z(x,y) + \eta \mathcal{U}(x,y). \tag{P}$$

**Proposition 7** (Regression model, squared loss). *Let $L$ be the squared loss. Then, for any $k \in [p]$, $\mathcal{L}(\mathcal{D}_k, h_z^\eta) - \mathcal{L}(\mathcal{D}_z, h_z^\eta) = u_k(z) - v_k(z)$, where $u_k$ and $v_k$ are convex functions defined for all $z$ by*

$$u_k(z) = \mathcal{L}\left(\mathcal{D}_k + \eta \mathcal{U}^1 \mathcal{D}_k(\cdot|x), h_z^\eta\right) - 2M \sum_x (\mathcal{D}_k^1 + \eta \mathcal{U}^1)(x) \log K_z(x),$$

$$v_k(z) = \mathcal{L}\left(\mathcal{D}_z + \eta \mathcal{U}^1 \mathcal{D}_k(\cdot|x), h_z^\eta\right) - 2M \sum_x (\mathcal{D}_k^1 + \eta \mathcal{U}^1)(x) \log K_z(x).$$

**Proposition 8** (Probability model, cross-entropy loss). *Let $L$ be the cross-entropy loss. Then, for $k \in [p]$, $\mathcal{L}(\mathcal{D}_k, h_z^\eta) - \mathcal{L}(\mathcal{D}_z, h_z^\eta) = u_k(z) - v_k(z)$, where $u_k$ and $v_k$ are convex functions defined for all $z$ by*

$$u_k(z) = -\sum_{x,y} \left[\mathcal{D}_k(x,y) + \eta \mathcal{U}(x,y)\right] \log J_z(x,y),$$

$$v_k(z) = \sum_{x,y} K_z(x,y) \log\left[\frac{K_z(x,y)}{J_z(x,y)}\right] - \left[\mathcal{D}_k(x,y) + \eta \mathcal{U}(x,y)\right] \log K_z(x,y).$$

## 4.3 DC algorithm

Our DC decompositions prove that the optimization problem (4) can be cast as the following variational form of a DC-programming problem (Tao and An, 1997, 1998; Sriperumbudur and Lanckriet, 2012):

$$\min_{z \in \Delta, \gamma \in \mathbb{R}} \gamma \quad \text{s.t.} \left(u_k(z) - v_k(z) \leq \gamma\right) \wedge \left(-z_k \leq 0\right) \wedge \left(\sum_{k=1}^{p} z_k - 1 = 0\right), \quad \forall k \in [p]. \tag{5}$$

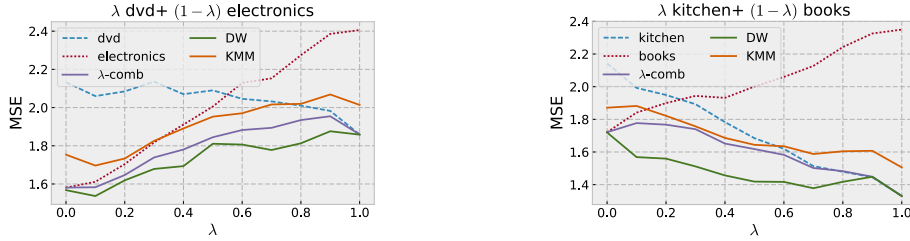

Figure 1: MSE sentiment analysis under mixture of two domains: (a) (*left figure*) `dvd` and `electronics`; (b) (*right figure*) `kitchen` and `books`.

The DC-programming algorithm works as follows. Let $(z_t)_t$ be the sequence defined by repeatedly solving the following convex optimization problem:

$$z_{t+1} \in \operatorname*{argmin}_{z, \gamma \in \mathbb{R}} \gamma \tag{6}$$

$$\text{s.t. } \big(u_k(z) - v_k(z_t) - (z - z_t)\nabla v_k(z_t) \le \gamma\big) \wedge \big(-z_k \le 0\big) \wedge \big(\textstyle\sum_{k=1}^{p} z_k - 1 = 0\big), \quad \forall k \in [p],$$

where $z_0 \in \Delta$ is an arbitrary starting value. Then, $(z_t)_t$ is guaranteed to converge to a local minimum of Problem (4) (Yuille and Rangarajan, 2003; Sriperumbudur and Lanckriet, 2012). Note that Problem (6) is a relatively simple optimization problem: $u_k(z)$ is a weighted sum of the negative logarithm of an affine function of $z$, plus a weighted sum of rational functions of $z$ (squared loss), and all other terms appearing in the constraints are affine functions of $z$.

Problem (4) seeks a parameter $z$ verifying $\mathcal{L}(\mathcal{D}_k, h_z^\eta) - \mathcal{L}(\mathcal{D}_z, h_z^\eta) \le \gamma$, for all $k \in [p]$ for an arbitrarily small value of $\gamma$. Since $\mathcal{L}(\mathcal{D}_z, h_z^\eta) = \sum_{k=1}^{p} z_k \mathcal{L}(\mathcal{D}_k, h_z^\eta)$ is a weighted average of the expected losses $\mathcal{L}(\mathcal{D}_k, h_z^\eta)$, $k \in [p]$, the solution $\gamma$ cannot be negative. Furthermore, by Lemma 6, a parameter $z$ verifying that inequality exists for any $\gamma > 0$. Thus, the global solution $\gamma$ of Problem (4) must be close to zero. This provides us with a simple criterion for testing the global optimality of the solution $z$ we obtain using a DC-programming algorithm with a starting parameter $z_0$.

## 5 Experiments

This section reports the results of our experiments with our DC-programming algorithm for finding a robust domain generalization solution when using squared loss and cross-entropy loss. We first evaluated our algorithm using an artificial dataset assuming known densities where we could compare our result to the global solution and found that indeed our global objective approached the known optimum of zero (see Appendix E for more details). Next, we evaluated our DC-programming solution applied to real-world datasets: a sentiment analysis dataset (Blitzer et al., 2007) with the squared loss, a visual domain adaptation benchmark dataset *Office* (Saenko et al., 2010), as well as a generalization of digit recognition task, with the cross-entropy loss.

For all real-world datasets, the probability distributions $\mathcal{D}_k$ are not readily available to the learner. However, Corollary 4 extends the learning guarantees of our solution to the case where an estimate $\widehat{\mathcal{D}}_k$ is used in lieu of the ideal distribution $\mathcal{D}_k$. Thus, we used standard density estimation methods to derive an estimate $\widehat{\mathcal{D}}_k$ for each $k \in [p]$. While density estimation can be a difficult task in general, for our purpose, straightforward techniques were sufficient for our predictor $\widehat{h}_z^\eta$ to achieve a high performance, since the approximate densities only serve to indicate the relative importance of each source domain. We give full details about our density estimation procedure in Appendix E.

### 5.1 Sentiment analysis task with the squared loss

We used the sentiment analysis dataset proposed by Blitzer et al. (2007) and used for multiple-source adaptation by Mansour et al. (2008, 2009a). This dataset consists of product review text and rating labels taken from four domains: `books` (B), `dvd` (D), `electronics` (E), and `kitchen` (K), with 2,000 samples for each domain. We defined a vocabulary of 2,500 words that occur at least twice in the intersection of the four domains. These words were used to define feature vectors, where every sample was encoded by the number of occurrences of each word. We trained our base hypotheses using support vector regression with the same hyper-parameters as in (Mansour et al., 2008, 2009a).

Table 1: MSE on the sentiment analysis dataset of source-only baselines for each domain, K,D, B,E, the uniform weighted predictor `unif`, KMM, and the distribution-weighted method DW based on the learned $z$. DW outperforms all competing baselines.

| | K | D | B | E | KD | BE | DBE | KBE | KDB | KDB | KDBE |
|---|---|---|---|---|---|---|---|---|---|---|---|
| | | | | | | Test Data | | | | | |
| K | 1.46±0.08 | 2.20±0.14 | 2.29±0.13 | 1.69±0.12 | 1.83±0.08 | 1.99±0.10 | 2.06±0.07 | 1.81±0.07 | 1.78±0.07 | 1.98±0.06 | 1.91±0.06 |
| D | 2.12±0.08 | 1.78±0.08 | 2.12±0.08 | 2.10±0.07 | 1.95±0.07 | 2.11±0.07 | 2.00±0.06 | 2.11±0.06 | 2.00±0.06 | 2.01±0.06 | 2.03±0.06 |
| B | 2.18±0.11 | 2.01±0.09 | 1.73±0.12 | 2.24±0.07 | 2.10±0.09 | 1.99±0.08 | 1.99±0.05 | 2.05±0.06 | 2.14±0.06 | 1.98±0.06 | 2.04±0.05 |
| E | 1.69±0.09 | 2.31±0.12 | 2.40±0.11 | 1.50±0.06 | 2.00±0.09 | 1.95±0.07 | 2.07±0.06 | 1.86±0.04 | 1.84±0.06 | 2.14±0.06 | 1.98±0.05 |
| unif | 1.62±0.05 | 1.84±0.09 | 1.86±0.09 | 1.62±0.07 | 1.73±0.06 | 1.74±0.07 | 1.77±0.05 | 1.70±0.05 | 1.69±0.04 | 1.77±0.04 | 1.74±0.04 |
| KMM | 1.63±0.15 | 2.07±0.12 | 1.93±0.17 | 1.69±0.12 | 1.83±0.07 | 1.82±0.07 | 1.89±0.07 | 1.75±0.07 | 1.78±0.06 | 1.86±0.09 | 1.82±0.06 |
| DW(ours) | **1.45±0.08** | **1.78±0.08** | **1.72±0.12** | **1.49±0.06** | **1.62±0.07** | **1.61±0.08** | **1.66±0.05** | **1.56±0.04** | **1.58±0.05** | **1.65±0.04** | **1.61±0.04** |

Table 2: **Digit** dataset statistics.

| | SVHN | MNIST | USPS |
|---|---|---|---|
| | 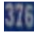 |  |  |
| # train images | 73,257 | 60,000 | 7,291 |
| # test images | 26,032 | 10,000 | 2,007 |
| image size | 32x32 | 28x28 | 16x16 |
| color | rgb | gray | gray |

Table 3: **Digit** dataset accuracy.

| | svhn | mnist | usps | mu | su | sm | smu | mean |
|---|---|---|---|---|---|---|---|---|
| | | | | Test Data | | | | |
| CNN-s | **92.3** | 66.9 | 65.6 | 66.7 | 90.4 | 85.2 | 84.2 | 78.8 |
| CNN-m | 15.7 | **99.2** | 79.7 | 96.0 | 20.3 | 38.9 | 41.0 | 55.8 |
| CNN-u | 16.7 | 62.3 | **96.6** | 68.1 | 22.5 | 29.4 | 32.9 | 46.9 |
| CNN-unif | 75.7 | 91.3 | 92.2 | 91.4 | 76.9 | 80.0 | 80.7 | 84.0 |
| DW (ours) | 91.4 | 98.8 | 95.6 | 98.3 | **91.7** | **93.5** | **93.6** | **94.7** |
| CNN-joint | 90.9 | 99.1 | 96.0 | **98.6** | 91.3 | 93.2 | 93.3 | 94.6 |

We compared our method (DW) against each source hypothesis, $h_k$. We also computed a privileged baseline using the oracle $\lambda$ mixing parameter, $\lambda$-comb: $\sum_{k=1}^{p} \lambda_k h_k$. $\lambda$-comb is of course not accessible in practice since the target mixture $\lambda$ is not known to the user. We also compared against a previously proposed domain adaptation algorithm (Huang et al., 2006) known as KMM. It is important to note that the KMM model requires access to the unlabeled target data during adaptation and learns a new predictor for every target domain, while DW does not use any target data. Thus KMM operates in a favorable learning setting when compared to our solution.

We first considered the same test scenario as in (Mansour et al., 2008), where the target is a mixture of two source domains. The plots of Figures 1a and 1b report the results of our experiments. They show that our distribution-weighted predictor DW outperforms all baseline predictors despite the privileged learning scenarios of $\lambda$-comb and KMM. We also compared our results with the *weighted predictor* used in the empirical studies by Mansour et al. (2008), which is not a realistic solution since it is using the unknown target mixture $\lambda$ as $z$ to compute $h_z$. Nevertheless, we observed that the performance of this "cheating" solution almost always coincides with that of our DW algorithm and thus did not include it in our plots and tables to avoid confusion.

Next, we compared the performance of DW with accessible baseline predictors on various target mixtures. Since $\lambda$ is not known in practice, we replaced $\lambda$-comb with the uniform combination of all hypotheses (unif), $\sum_{k=1}^{p} h_k/p$. Table 1 reports the mean and standard deviations of MSE over 10 repetitions. Each column corresponds to a different target test data source. Our distribution-weighted method DW outperforms all baseline predictors across all test domains. Observe that, even when the target is a single source domain, our method successfully outperforms the predictor which is trained and tested on the same domain. Results on more target mixtures are available in Appendix E.

## 5.2 Recognition tasks with the cross-entropy loss

We considered two real-world domain adaptation tasks: a generalization of a digit recognition task and a standard visual adaptation *Office* dataset.

For each individual domain, we trained a convolutional neural network (CNN) and used the output from the softmax score layer as our base predictors $h_k$. We computed the uniformly weighted combination of source predictors, $h_{\text{unif}} = \sum_{k=1}^{p} h_k/p$. As a privileged baseline, we also trained a model on all source data combined, $h_{\text{joint}}$. Note, this approach is often not feasible if independent entities contribute classifiers and densities, but not full training datasets. Thus this approach is not consistent with our scenario, and it operates in a much more favorable learning setting than our solution. Finally, our distribution weighted predictor DW was computed with $h_k$s, density estimates, and our learned weighting, $z$. Our baselines then consists of the classifiers from $h_k$, $h_{\text{unif}}$, $h_{\text{joint}}$, and DW.

Table 4: ***Office*** dataset accuracy: We report accuracy across six possible test domains. We show performance all baselines: CNN-a,w,d, CNN-unif, DW based on the learned $z$, and the jointly trained model CNN-joint. DW outperforms all competing models.

| | amazon | webcam | dslr | aw | ad | wd | awd | mean |
|---|---|---|---|---|---|---|---|---|
| | | | | Test Data | | | | |
| CNN-a | **75.7 ± 0.3** | 53.8 ± 0.7 | 53.4 ± 1.3 | 71.4 ± 0.3 | 73.5 ± 0.2 | 53.6 ± 0.8 | 69.9 ± 0.3 | 64.5 ± 0.6 |
| CNN-w | 45.3 ± 0.5 | 91.1 ± 0.8 | 91.7 ± 1.2 | 54.4 ± 0.5 | 50.0 ± 0.5 | 91.3 ± 0.8 | 57.5 ± 0.4 | 68.8 ± 0.7 |
| CNN-d | 50.4 ± 0.4 | 89.6 ± 0.9 | 90.9 ± 0.8 | 58.3 ± 0.4 | 54.6 ± 0.4 | 90.0 ± 0.7 | 61.0 ± 0.4 | 70.7 ± 0.6 |
| CNN-unif | 69.7 ± 0.3 | 93.1 ± 0.6 | 93.2 ± 0.9 | 74.4 ± 0.4 | 72.1 ± 0.3 | 93.1 ± 0.5 | 75.9 ± 0.3 | 81.6 ± 0.5 |
| DW (ours) | 75.2 ± 0.4 | **93.7 ± 0.6** | **94.0 ± 1.0** | **78.9 ± 0.4** | **77.2 ± 0.4** | **93.8 ± 0.6** | **80.2 ± 0.3** | **84.7 ± 0.5** |
| CNN-joint | 72.1 ± 0.3 | 93.7 ± 0.5 | 93.7 ± 0.5 | 76.4 ± 0.4 | 76.4 ± 0.4 | 93.7 ± 0.5 | 79.3 ± 0.4 | 83.6 ± 0.4 |

We began our study with a generalization of digit recognition task, which consists of three digit recognition datasets: Google Street View House Numbers (SVHN), MNIST, and USPS. Dataset statistics as well as example images can be found in Table 2. We trained the ConvNet (or CNN) architecture following Taigman et al. (2017) as our source models and joint model. We used the second fully-connected layer's output as our features for density estimation, and the output from the softmax score layer as our predictors. We used the full training sets per domain to learn the source model and densities. Note, these steps are completely isolated from one another and may be performed by unique entities and in parallel. Finally, for our DC-programming algorithm we used a small subset of 200 real image-label pairs from each domain to learn the parameter $z$.

Our next experiment used the standard visual adaptation *Office* dataset, which has 3 domains: amazon, webcam, and dslr. The dataset contains 31 recognition categories of objects commonly found in an office environment. There are $4,110$ images total with $2,817$ from amazon, $795$ from webcam, and $498$ from dslr.

We followed the standard protocol from Saenko et al. (2010), whereby 20 labeled examples are available for training from the amazon domain and 8 labeled examples are available from both the webcam and dslr domains. The remaining examples from each domain are used for testing. We used the AlexNet Krizhevsky et al. (2012) ConvNet (CNN) architecture, and used the output from the softmax score layer as our base predictors, pre-trained on ImageNet and used fc7 activations as our features for density estimation Donahue et al. (2014).

We report the performance of our algorithm and that of baselines on the digit recognition dataset in Table 3, and report the performance on the *Office* dataset in Table 4. On both datasets, we evaluated on various test distributions: each individual domain, the combination of each two domains and the fully combined set. When the test distribution equals one of the source distributions, our distribution-weighted classifier successfully outperforms (webcam,dslr) or maintains the performance of the classifier which is trained and tested on the same domain. For the more realistic scenario where the target domain is a mixture of any two or all three source domains, the performance of our method is comparable or marginally superior to that of the jointly trained network, despite the fact that we do not retrain any network parameters in our method and that we only use a small number of per-domain examples to learn the distribution weights – an optimization which may be solved on a single CPU in a matter of seconds for this problem. This again demonstrates the robustness of our distribution-weighted combined classifier to a varying target domain.

## 6   Conclusion

We presented practically applicable multiple-source domain adaptation algorithms for the squared loss and the cross-entropy loss. Our algorithms benefit from a series of very favorable theoretical guarantees. Our results further demonstrate empirically their effectiveness and their importance in adaptation problems in practice.

**Acknowledgments**

We thank Cyril Allauzen for comments on a previous draft of this paper. This work was partly funded by NSF CCF-1535987 and NSF IIS-1618662.

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
