[Supplementary Material]

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

## Footnotes

[1] The unnormalized relative entropy of $P$ and $Q$ is defined by $B(P \parallel Q) = \sum_{x,y} P(x,y)\log\left[\frac{P(x,y)}{Q(x,y)}\right] + \sum_{(x,y)}(Q(x,y) - P(x,y))$.

[2] To be precise, it can be shown that the relative entropy is jointly convex using the so-called log-sum inequality (Cover and Thomas, 2006). The same proof using the log-sum inequality can be used to show the joint convexity of the unnormalized relative entropy.

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

# A   Lower bounds for convex combination rules

In this section, we give lower bounds for convex combination rule, for both squared loss and cross-entropy loss. For any $\alpha \in \Delta$, we define the convex combination rule for the regression and the probability model as follows:

$$g_\alpha(x) = \sum_{k=1}^{p} \alpha_k h_k(x), \qquad\qquad (R) \qquad\qquad (7)$$

$$g_\alpha(x,y) = \sum_{k=1}^{p} \alpha_k h_k(x,y). \qquad\qquad (P) \qquad\qquad (8)$$

**Lemma 9** (*Regression model*, squared loss)**.** *There is a mixture adaptation problem for which the expected squared loss of $g_\alpha$ is $\frac{1}{4}$.*

*Proof.* Let $\mathcal{X} = \{a,b\}$, and $\mathcal{Y} = \{0,1\}$. Consider $\mathcal{D}_0(x,y) = 1_{x=a,y=0}$, $h_0(x) = 0$, and $\mathcal{D}_1(x,y) = 1_{x=b,y=1}$, $h_1(x) = 1$. Consider the target distribution $\mathcal{D}_T = \frac{1}{2}\mathcal{D}_0 + \frac{1}{2}\mathcal{D}_1$. Then, for any convex combination rule $g_\alpha = \alpha h_0 + (1-\alpha)h_1 = 1 - \alpha$,

$$\left(\frac{1}{2}\right)^2 = \left(\frac{1}{2}\alpha + \frac{1}{2}(1-\alpha)\right)^2 = \left(\sum_{(x,y)\in\mathcal{X}\times\mathcal{Y}} \mathcal{D}_T(x,y)|g_\alpha(x) - y|\right)^2$$

$$\leq \sum_{(x,y)\in\mathcal{X}\times\mathcal{Y}} \mathcal{D}_T(x,y)(g_\alpha(x) - y)^2 = \mathcal{L}(\mathcal{D}_T, g_\alpha).$$

$\square$

Note that the hypotheses $h_0$ and $h_1$ have *zero* error on their own domain, i.e. $\epsilon = 0$. However, *no* convex combination rule will perform well on the target distribution $\mathcal{D}_T$.

**Lemma 10** (*Probability Model*, cross-entropy loss)**.** *There is a mixture adaptation problem for which the expected cross-entropy loss of $g_\alpha$ is $\log(p)$.*

*Proof.* Let $\mathcal{X} = \{x_1, \ldots, x_k\}$, and $\mathcal{Y} = \{y_1, \ldots, y_k\}$. Consider $\mathcal{D}_k(x,y) = 1_{x=x_k,y=y_k}$, and $h_k(x,y) = 1_{y=y_k}$. Consider the largest cross-entropy loss of $g_\alpha$ on any target mixture $\mathcal{D}_\lambda(x,y)$:

$$\max_{\lambda\in\Delta}\mathcal{L}(\mathcal{D}_\lambda, g_\alpha) = \max_{\lambda\in\Delta}\sum_{k=1}^{p} -\lambda_k \log(\alpha_k) = \max_{k\in[p]}\left[-\log(\alpha_k)\right].$$

Choosing $\alpha \in \Delta$ to minimize that adversarial loss gives

$$\min_{\alpha\in\Delta}\max_{k\in[p]}\left[-\log(\alpha_k)\right] = \log(p).$$

Therefore any convex combination rule $g_\alpha$ incurs at least a loss of $\log(p)$. $\square$

Again, the base hypotheses $h_k$s have *zero* error on their own domain, yet there is no convex combination rule that is robust against arbitrary target mixture.

# B Theoretical analysis for the stochastic scenario

In this section, we give a series of theoretical results for the general stochastic scenario with their full proofs. We will separate the proofs for the regression model (Appendix B.1) and the probability model (Appendix B.3), since the definitions of the distribution weighted combination are different in the two models.

## B.1 Regression model

The proofs for the regression model (R) are presented in the following order: we first assume the conditional probabilities are the same across source domains, and prove Lemma 6; using that, we prove Corollary 3 and Corollary 4. Finally, we relax the assumption of same conditionals, and prove Theorem 2, which is a stronger version of Theorem 1.

Our proofs make use of the following Fixed-Point Theorem of Brouwer.

**Theorem 11.** *For any compact and convex non-empty set $C \subset \mathbb{R}^p$ and any continuous function $f: C \to C$, there is a point $x \in C$ such that $f(x) = x$.*

**Lemma 6.** *For any $\eta, \eta' > 0$, there exists $z \in \Delta$, with $z_k \neq 0$ for all $k \in [p]$, such that the following holds for the distribution-weighted combining rule $h_z^\eta$:*

$$\forall k \in [p], \quad \mathcal{L}(\mathcal{D}_k, h_z^\eta) \leq \sum_{j=1}^p z_j \mathcal{L}(\mathcal{D}_j, h_z^\eta) + \eta'. \tag{9}$$

*Proof.* Consider the mapping $\Phi: \Delta \to \Delta$ defined for all $z \in \Delta$ by

$$[\Phi(z)]_k = \frac{z_k \, \mathcal{L}(\mathcal{D}_k, h_z^\eta) + \frac{\eta'}{p}}{\sum_{j=1}^p z_j \mathcal{L}(\mathcal{D}_j, h_z^\eta) + \eta'}.$$

$\Phi$ is continuous since $\mathcal{L}(\mathcal{D}_k, h_z^\eta)$ is a continuous function of $z$ and since the denominator is positive ($\eta' > 0$). Thus, by Brouwer's Fixed Point Theorem, there exists $z \in \Delta$ such that $\Phi(z) = z$. For that $z$, we can write

$$z_k = \frac{z_k \, \mathcal{L}(\mathcal{D}_k, h_z^\eta) + \frac{\eta'}{p}}{\sum_{j=1}^p z_j \mathcal{L}(\mathcal{D}_j, h_z^\eta) + \eta'},$$

for all $k \in [p]$. Since $\eta'$ is positive, we must have $z_k \neq 0$ for all $k$. Dividing both sides by $z_k$ gives $\mathcal{L}(\mathcal{D}_k, h_z^\eta) = \sum_{j=1}^p z_j \mathcal{L}(\mathcal{D}_j, h_z^\eta) + \eta' - \frac{\eta'}{p z_k} \leq \sum_{j=1}^p z_j \mathcal{L}(\mathcal{D}_j, h_z^\eta) + \eta'$, which completes the proof. □

**Corollary 3.** *Assume that the conditional probability distributions $\mathcal{D}_k(\cdot|x)$ do not depend on $k$. Then, for any $\delta > 0$, there exist $\eta > 0$ and $z \in \Delta$ such that $\mathcal{L}(\mathcal{D}_\lambda, h_z^\eta) \leq \epsilon + \delta$ for any mixture parameter $\lambda \in \Delta$.*

*Proof.* We first upper bound, for an arbitrary $z \in \Delta$, the expected loss of $h_z^\eta$ with respect to the mixture distribution $\mathcal{D}_z$ defined using the same $z$, that is $\mathcal{L}(\mathcal{D}_z, h_z^\eta) = \sum_{k=1}^p z_k \mathcal{L}(\mathcal{D}_k, h_z^\eta)$. By definition of $h_z^\eta$ and $\mathcal{D}_z$, we can write

$$\mathcal{L}(\mathcal{D}_z, h_z^\eta) = \sum_{(x,y)} \mathcal{D}_z(x,y) L(h_z^\eta(x), y)$$

$$= \sum_{(x,y)} \mathcal{D}_z(x,y) L\left(\sum_{k=1}^p \frac{z_k \mathcal{D}_k^1(x) + \eta \frac{\mathcal{U}^1(x)}{p}}{\mathcal{D}_z^1(x) + \eta \mathcal{U}^1(x)} h_k(x), y\right).$$

By convexity of $L$, this implies that

$$\mathcal{L}(\mathcal{D}_z, h_z^{\eta}) \leq \sum_{(x,y)} \mathcal{D}_z(x,y) \sum_{k=1}^p \frac{z_k \mathcal{D}_k^1(x) + \eta \frac{\mathcal{U}^1(x)}{p}}{\mathcal{D}_z^1(x) + \eta \mathcal{U}^1(x)} L(h_k(x), y)$$

$$\leq \sum_{(x,y)} \mathcal{D}_z(y|x) \mathcal{D}_z^1(x) \sum_{k=1}^p \frac{z_k \mathcal{D}_k^1(x) + \eta \frac{\mathcal{U}^1(x)}{p}}{\mathcal{D}_z^1(x) + \eta \mathcal{U}^1(x)} L(h_k(x), y)$$

$$\leq \sum_{(x,y)} \mathcal{D}_z(y|x) \sum_{k=1}^p \left( z_k \mathcal{D}_k^1(x) + \eta \frac{\mathcal{U}^1(x)}{p} \right) L(h_k(x), y).$$

Next, observe that $\mathcal{D}_z(y|x) = \sum_{k=1}^p \frac{z_k \mathcal{D}_k^1(x)}{\mathcal{D}_z^1(x)} \mathcal{D}_k(y|x) = \mathcal{D}_k(y|x)$ for any $k \in [p]$ since by assumption $\mathcal{D}_k(y|x)$ does not depend on $k$. Thus,

$$\mathcal{L}(\mathcal{D}_z, h_z^{\eta}) \leq \sum_{(x,y)} \mathcal{D}_z(y|x) \sum_{k=1}^p \left( z_k \mathcal{D}_k^1(x) + \eta \frac{\mathcal{U}^1(x)}{p} \right) L(h_k(x), y)$$

$$= \sum_{(x,y)} \sum_{k=1}^p \left( z_k \mathcal{D}_k(x,y) + \eta \mathcal{D}_k(y|x) \frac{\mathcal{U}^1(x)}{p} \right) L(h_k(x), y)$$

$$= \sum_{k=1}^p z_k \mathcal{L}(\mathcal{D}_k, h_k) + \frac{\eta}{p} \sum_{k=1}^p \sum_{(x,y)} \mathcal{D}_k(y|x) \mathcal{U}^1(x) L(h_k(x), y)$$

$$\leq \sum_{k=1}^p z_k \mathcal{L}(\mathcal{D}_k, h_k) + \eta M \leq \sum_{k=1}^p z_k \epsilon + \eta M = \epsilon + \eta M.$$

Now, choose $z \in \Delta$ as in the statement of Lemma 6. Then, the following holds for any mixture distribution $\mathcal{D}_\lambda$:

$$\mathcal{L}(\mathcal{D}_\lambda, h_z^{\eta}) = \sum_{k=1}^p \lambda_k \mathcal{L}(\mathcal{D}_k, h_z^{\eta}) \leq \sum_{k=1}^p \lambda_k \left( \mathcal{L}(\mathcal{D}_z, h_z^{\eta}) + \eta' \right)$$

$$= \mathcal{L}(\mathcal{D}_z, h_z^{\eta}) + \eta' \leq \epsilon + \eta M + \eta'.$$

Setting $\eta = \frac{\delta}{2M}$ and $\eta' = \frac{\delta}{2}$ concludes the proof. $\qquad\square$

Next, we extend to the case where the target distribution is arbitrary, that is, the target distribution is not necessarily a mixture of source distributions.

**Corollary 12.** *For any $\delta > 0$, there exist $\eta > 0$ and $z \in \Delta$, such that the following inequality holds for any $\alpha > 1$ and arbitrary target distribution $\mathcal{D}_T$:*

$$\mathcal{L}(\mathcal{D}_T, h_z^{\eta}) \leq \left[ (\epsilon + \delta) \, \mathsf{d}_\alpha(\mathcal{D}_T \parallel \mathcal{D}) \right]^{\frac{\alpha-1}{\alpha}} M^{\frac{1}{\alpha}}.$$

*Proof.* For any hypothesis $h \colon \mathcal{X} \to \mathcal{Y}$ and any distribution $\mathcal{D}$, by Hölder's inequality, the following holds:

$$\mathcal{L}(\mathcal{D}_T, h) = \sum_{(x,y)\in\mathcal{X}\times\mathcal{Y}} \mathcal{D}_T(x,y) L(h(x), y)$$

$$= \sum_{(x,y)\in\mathcal{X}\times\mathcal{Y}} \left[ \frac{\mathcal{D}_T(x,y)}{\mathcal{D}(x,y)^{\frac{\alpha-1}{\alpha}}} \right] \left[ \mathcal{D}(x,y)^{\frac{\alpha-1}{\alpha}} L(h(x), y) \right]$$

$$\leq \left[ \sum_{(x,y)} \frac{\mathcal{D}_T(x,y)^{\alpha}}{\mathcal{D}(x,y)^{\alpha-1}} \right]^{\frac{1}{\alpha}} \left[ \sum_{(x,y)} \mathcal{D}(x,y) L(h(x), y)^{\frac{\alpha}{\alpha-1}} \right]^{\frac{\alpha-1}{\alpha}}.$$

Thus, by definition of $\mathsf{d}_\alpha$, for any $h$ such that $L(h(x), y) \leq M$ for all $(x, y)$, we can write

$$\mathcal{L}(\mathcal{D}_T, h) \leq \mathsf{d}_\alpha(\mathcal{D}_T \parallel \mathcal{D})^{\frac{\alpha-1}{\alpha}} \left[ \sum_{(x,y)} \mathcal{D}(x,y) L(h(x),y)^{\frac{\alpha}{\alpha-1}} \right]^{\frac{\alpha-1}{\alpha}}$$

$$= \mathsf{d}_\alpha(\mathcal{D}_T \parallel \mathcal{D})^{\frac{\alpha-1}{\alpha}} \left[ \sum_{(x,y)} \mathcal{D}(x,y) L(h(x),y) L(h(x),y)^{\frac{1}{\alpha-1}} \right]^{\frac{\alpha-1}{\alpha}}$$

$$\leq \mathsf{d}_\alpha(\mathcal{D}_T \parallel \mathcal{D})^{\frac{\alpha-1}{\alpha}} \left[ \sum_{(x,y)} \mathcal{D}(x,y) L(h(x),y) M^{\frac{1}{\alpha-1}} \right]^{\frac{\alpha-1}{\alpha}}$$

$$\leq \left[ \mathsf{d}_\alpha(\mathcal{D}_T \parallel \mathcal{D}) \, \mathcal{L}(\mathcal{D}, h) \right]^{\frac{\alpha-1}{\alpha}} M^{\frac{1}{\alpha}}.$$

Now, by Corollary 3, there exist $z \in \Delta$ and $\eta > 0$ such that $\mathcal{L}(\mathcal{D}, h_z^\eta) \leq \epsilon + \delta$ for any mixture distribution $\mathcal{D} \in \mathcal{D}$. Thus, in view of the previous inequality, we can write, for any $\mathcal{D} \in \mathcal{D}$,

$$\mathcal{L}(\mathcal{D}_T, h_z^\eta) \leq \left[ (\epsilon + \delta) \, \mathsf{d}_\alpha(\mathcal{D}_T \parallel \mathcal{D}) \right]^{\frac{\alpha-1}{\alpha}} M^{\frac{1}{\alpha}}.$$

Taking the infimum of the right-hand side over all $\mathcal{D} \in \mathcal{D}$ completes the proof. $\qquad\square$

**Corollary 4.** *For any $\delta > 0$, there exist $\eta > 0$ and $z \in \Delta$, such that the following inequality holds for any $\alpha > 1$ and arbitrary target distribution $\mathcal{D}_T$:*

$$\mathcal{L}(\mathcal{D}_T, \widehat{h}_z^\eta) \leq \left[ (\widehat{\epsilon} + \delta) \, \mathsf{d}_\alpha(\mathcal{D}_T \parallel \widehat{\mathcal{D}}) \right]^{\frac{\alpha-1}{\alpha}} M^{\frac{1}{\alpha}},$$

*where $\widehat{\epsilon} = \max_{k \in [p]} \left[ \epsilon \, \mathsf{d}_\alpha(\widehat{\mathcal{D}}_k \parallel \mathcal{D}_k) \right]^{\frac{\alpha-1}{\alpha}} M^{\frac{1}{\alpha}}$, and $\widehat{\mathcal{D}} = \left\{ \sum_{k=1}^p \lambda_k \widehat{\mathcal{D}}_k : \lambda \in \Delta \right\}$.*

*Proof.* By the first part of the proof of Corollary 12, for any $k \in [p]$ and $\alpha > 1$, the following inequality holds:

$$\mathcal{L}(\widehat{\mathcal{D}}_k, h_k) \leq \left[ \mathsf{d}_\alpha(\widehat{\mathcal{D}}_k \parallel \mathcal{D}_k) \, \mathcal{L}(\mathcal{D}_k, h_k) \right]^{\frac{\alpha-1}{\alpha}} M^{\frac{1}{\alpha}}$$

$$\leq \left[ \epsilon \, \mathsf{d}_\alpha(\widehat{\mathcal{D}}_k \parallel \mathcal{D}_k) \right]^{\frac{\alpha-1}{\alpha}} M^{\frac{1}{\alpha}} \leq \widehat{\epsilon}.$$

We can now apply the result of Corollary 12 (with $\widehat{\epsilon}$ instead of $\epsilon$ and $\widehat{\mathcal{D}}_k$ instead of $\mathcal{D}_k$). In view that, there exist $\eta > 0$ and $z \in \Delta$ such that

$$\mathcal{L}(\mathcal{D}_T, \widehat{h}_z^\eta) \leq \left[ (\widehat{\epsilon} + \delta) \, \mathsf{d}_\alpha(\mathcal{D}_T \parallel \widehat{\mathcal{D}}) \right]^{\frac{\alpha-1}{\alpha}} M^{\frac{1}{\alpha}},$$

for any distribution $\widehat{\mathcal{D}}$ in the family $\widehat{\mathcal{D}}$. Taking the infimum over all $\widehat{\mathcal{D}}$ in $\widehat{\mathcal{D}}$ completes the proof. $\quad\square$

Corollary 4 uses Rényi divergence in both directions: $\mathsf{d}_\alpha(\mathcal{D}_T \parallel \widehat{\mathcal{D}})$ requires $\mathrm{Supp}(\mathcal{D}_T) \subseteq \mathrm{Supp}(\widehat{\mathcal{D}})$, and $\mathsf{d}_\alpha(\widehat{\mathcal{D}}_k \parallel \mathcal{D}_k)$ requires $\mathrm{Supp}(\widehat{\mathcal{D}}_k) \subseteq \mathrm{Supp}(\mathcal{D}_k)$, $k \in [p]$. In our experiments in Section 5, we used a bigram language model for sentiment analysis, and kernel density estimation with a Gaussian kernel for object recognition. Both density estimation methods fulfill these requirements.

Finally we prove our main result Theorem 1 under the regression model (R). We do so by proving a more general result, Theorem 2, and showing that it will coincide with Theorem 1 under the assumption that $\mathcal{D}_T^1 \in \mathcal{D}^1$ and $\mathcal{D}_T(\cdot|x)$ coincides for all $\mathcal{D}_T$.

**Theorem 2** (Regression model). *Fix a conditional probability distribution $\mathcal{Q}(\cdot|x)$ defined for all $x \in \mathcal{X}$. Then, for any $\delta > 0$, there exist $\eta > 0$ and $z \in \Delta$ such that the following inequality holds for any $\alpha, \beta > 1$ and any target distribution $\mathcal{D}_T$:*

$$\mathcal{L}(\mathcal{D}_T, h_z^\eta) \leq \left[ \left( \epsilon_\alpha(\mathcal{Q}) + \delta \right) \mathsf{d}_\beta(\mathcal{D}_T \parallel \mathcal{D}_{P,\mathcal{Q}}) \right]^{\frac{\beta-1}{\beta}} M^{\frac{1}{\beta}}. \tag{10}$$

*Proof.* For any $k \in [p]$, by Hölder's inequality, the following holds:

$$\mathcal{L}(\mathcal{D}_{k,\mathcal{Q}}, h_k) = \sum_{x,y} \mathcal{D}_k^1(x) \mathcal{Q}(y|x) L(h_k, x, y)$$

$$= \sum_x \mathcal{D}_k^1(x) \sum_y \left[ \frac{\mathcal{Q}(y|x)}{\mathcal{D}_k(y|x)^{\frac{\alpha-1}{\alpha}}} \right] \left[ \mathcal{D}_k(y|x)^{\frac{\alpha-1}{\alpha}} L(h_k, x, y) \right]$$

$$\leq \sum_x \mathcal{D}_k^1(x) \mathsf{d}_\alpha(x; \mathcal{Q}, k)^{\frac{\alpha-1}{\alpha}} \left[ \sum_y \mathcal{D}_k(y|x) L(h_k, x, y)^{\frac{\alpha}{\alpha-1}} \right]^{\frac{\alpha-1}{\alpha}},$$

where, for simplicity, we write $\mathsf{d}_\alpha(x; \mathcal{Q}, k) = \mathsf{d}_\alpha\left( \mathcal{Q}(\cdot|x) \parallel \mathcal{D}_k(\cdot|x) \right)$. Using the boundedness of the loss and Hölder's inequality again, we can write

$$\mathcal{L}(\mathcal{D}_{k,\mathcal{Q}}, h_k) \leq \sum_x \mathcal{D}_k^1(x)^{\frac{1}{\alpha}} \mathsf{d}_\alpha(x; \mathcal{Q}, k)^{\frac{\alpha-1}{\alpha}} \left[ \sum_y \mathcal{D}_k(x, y) L(h_k, x, y) \right]^{\frac{\alpha-1}{\alpha}} M^{\frac{1}{\alpha}}$$

$$\leq \left[ \sum_x \mathcal{D}_k^1(x) \mathsf{d}_\alpha(x; \mathcal{Q}, k)^{\alpha-1} \right]^{\frac{1}{\alpha}} \left[ \sum_{x,y} \mathcal{D}_k(x, y) L(h_k, x, y) \right]^{\frac{\alpha-1}{\alpha}} M^{\frac{1}{\alpha}}$$

$$\leq \left[ \mathop{\mathbb{E}}_{\mathcal{D}_k^1} \left[ \mathsf{d}_\alpha(x; \mathcal{Q}, k)^{\alpha-1} \right] \right]^{\frac{1}{\alpha}} \epsilon^{\frac{\alpha-1}{\alpha}} M^{\frac{1}{\alpha}} \leq \epsilon_\alpha(\mathcal{Q}).$$

We can now apply the result of Corollary 12, with $\beta$ instead of $\alpha$, $\epsilon_\alpha(\mathcal{Q})$ instead of $\epsilon$ and $\mathcal{D}_{k,\mathcal{Q}}$ instead of $\mathcal{D}_k$. This completes the proof. $\qquad\square$

When $\mathcal{D}_T^1 \in \mathcal{D}^1$, $\mathcal{D}_T^1(x)\mathcal{Q}(y|x) \in \mathcal{D}_{P,\mathcal{Q}}$ and we can write

$$\mathsf{d}_\beta(\mathcal{D}_T \parallel \mathcal{D}_{P,\mathcal{Q}}) \leq \left[ \sum_{x,y} \frac{[\mathcal{D}_T^1(x)\mathcal{D}_T(y|x)]^\beta}{[\mathcal{D}_T^1(x)\mathcal{Q}(y|x)]^{\beta-1}} \right]^{\frac{1}{\beta-1}}$$

$$= \left[ \sum_x \mathcal{D}_T^1(x) \sum_y \frac{[\mathcal{D}_T(y|x)]^\beta}{[\mathcal{Q}(y|x)]^{\beta-1}} \right]^{\frac{1}{\beta-1}}$$

$$= \left[ \mathop{\mathbb{E}}_{\mathcal{D}_T^1} \left[ \mathsf{d}_\beta\left( \mathcal{D}_T(\cdot|x) \parallel \mathcal{Q}(\cdot|x) \right)^{\beta-1} \right] \right]^{\frac{1}{\beta-1}}.$$

Applying this inequality to (10) yields

$$\mathcal{L}(\mathcal{D}_T, h_z^\eta) \leq \left[ \left( \epsilon_\alpha(\mathcal{Q}) + \delta \right) \mathsf{d}_\beta(\mathcal{D}_T \parallel \mathcal{D}_{P,\mathcal{Q}}) \right]^{\frac{\beta-1}{\beta}} M^{\frac{1}{\beta}}$$

$$\leq \left( \epsilon_\alpha(\mathcal{Q}) + \delta \right)^{\frac{\beta-1}{\beta}} \left[ \mathop{\mathbb{E}}_{\mathcal{D}_T^1} \left[ \mathsf{d}_\beta\left( \mathcal{D}_T(\cdot|x) \parallel \mathcal{Q}(\cdot|x) \right)^{\beta-1} \right] \right]^{\frac{1}{\beta}} M^{\frac{1}{\beta}}. \qquad (11)$$

Notice that when the target distribution $\mathcal{D}_T$ is arbitrary but admits a fixed (and unknown) conditional probability distribution $\mathcal{D}_T(\cdot|x)$, we can set $\mathcal{Q}(\cdot|x) = \mathcal{D}_T(\cdot|x)$ in (11). We then have $\mathsf{d}_\beta(\mathcal{D}_T(\cdot|x) \parallel \mathcal{Q}(\cdot|x)) = 1$ for all $x \in \mathcal{X}$ and $\mathcal{L}(\mathcal{D}_T, h_z^\eta) \leq \left( \epsilon_\alpha(\mathcal{Q}) + \delta \right)^{\frac{\beta-1}{\beta}} M^{\frac{1}{\beta}}$. Thus, Theorem 2 coincides with the statement of Theorem 1 for the regression model by setting $\beta = +\infty$.

## B.2  Choice of $z$

We have shown the existence of a robust solution $h_z^\eta$ that works well for arbitrary target distribution $\mathcal{D}_T$. However, in the proof of Theorem 2, the choice of $z$ depends on a fixed conditional probability distribution $\mathcal{Q}(\cdot|x)$. In practice, if the learner assumes that the conditional probability distribution $\mathcal{D}_k(\cdot|x)$ coincides, he can then set $\mathcal{Q}(\cdot|x) = \mathcal{D}_k(\cdot|x)$ and use $\mathcal{D}_k$ to solve the DC programming problem (4) for $z$. When the conditional probabilities are distinct, however, the learner needs to first come up with a choice of $\mathcal{Q}(\cdot|x)$, and then solve the DC programming problem (4) for $z$ with $\mathcal{D}_{k,\mathcal{Q}}$ instead of $\mathcal{D}_k$, and the theoretical guarantees of $h_z$ depend on $\mathcal{Q}(\cdot|x)$.

Can we find a robust solution $z$ using only the original distributions $\mathcal{D}_k, \forall k \in [p]$, even when the conditional probability distributions $\mathcal{D}_k(\cdot|x)$ vary by $k$? The answer is yes. We now prove a

variant of Theorem 2 where the choice of $z$ only depends on $\mathcal{D}_k, \forall k \in [p]$. This variant allows us to always use $\mathcal{D}_k$s in the DC programming formulation (4). In what follows, we denote by $\mathcal{D} = \{\sum_{k=1}^{p} \lambda_k \mathcal{D}_k, \lambda \in \Delta\}$, and $\mathcal{D}_{z,\mathcal{Q}}(x,y) = \sum_{k=1}^{p} z_k \mathcal{D}_{k,\mathcal{Q}}$.

**Theorem 13.** *Given any $\eta, \eta' > 0$, there exits $z \in \Delta$ such that the following holds for any $\lambda \in \Delta$:*

$$\mathcal{L}(\mathcal{D}_\lambda, h_z^\eta) \le \min_{\mathcal{Q}(\cdot|x)} \left\{ \left[ \mathsf{d}_\alpha(\mathcal{D}_z \parallel \mathcal{D}_{z,\mathcal{Q}}) \right]^{\frac{\alpha-1}{\alpha}} \left[ \max_{k \in [p]} \mathsf{d}_\alpha(\mathcal{D}_{k,\mathcal{Q}} \parallel \mathcal{D}_k) \right]^{\frac{(\alpha-1)^2}{\alpha^2}} M^{\frac{2\alpha-1}{\alpha^2}} \epsilon^{\frac{(\alpha-1)^2}{\alpha^2}} \right.$$

$$\left. + \left[ \mathsf{d}_\alpha(\mathcal{D}_z \parallel \mathcal{D}_{z,\mathcal{Q}}) \right]^{\frac{\alpha-1}{\alpha}} M \eta^{\frac{\alpha-1}{\alpha}} + \eta' \right\}. \tag{12}$$

*When the conditional probability distributions $\mathcal{D}_k(\cdot|x)$ do not depend on $k$, (12) recovers the result of Corollary 3.*

*Furthermore, denote by $\mathcal{E}(\epsilon, \alpha, \eta, \eta')$ the above upper bound of $\mathcal{L}(\mathcal{D}_\lambda, h_z^\eta)$ for any $\lambda \in \Delta$. There exists $z \in \Delta$ such that for arbitrary target distribution $\mathcal{D}_T$,*

$$\mathcal{L}(\mathcal{D}_T, h_z^\eta) \le \left[ \mathcal{E}(\epsilon, \alpha, \eta, \eta') \, \mathsf{d}_\alpha(\mathcal{D}_T \parallel \mathcal{D}) \right]^{\frac{\alpha-1}{\alpha}} M^{\frac{1}{\alpha}}.$$

*Proof.* Given any conditional probability distribution $\mathcal{Q}(\cdot|x)$, by the proof of Corollary 12, for any $z \in \Delta$,

$$\mathcal{L}(\mathcal{D}_z, h_z^\eta) \le \left[ \mathsf{d}_\alpha(\mathcal{D}_z \parallel \mathcal{D}_{z,\mathcal{Q}}) \, \mathcal{L}(\mathcal{D}_{z,\mathcal{Q}}, h_z^\eta) \right]^{\frac{\alpha-1}{\alpha}} M^{\frac{1}{\alpha}}. \tag{13}$$

By the proof of Corollary 3 and Corollary 12,

$$\mathcal{L}(\mathcal{D}_{z,\mathcal{Q}}, h_z^\eta) \le \sum_{k=1}^{p} z_k \mathcal{L}(\mathcal{D}_{k,\mathcal{Q}}, h_k) + \eta M$$

$$\le \sum_{k=1}^{p} z_k \left[ \mathsf{d}_\alpha(\mathcal{D}_{k,\mathcal{Q}} \parallel \mathcal{D}_k) \, \mathcal{L}(\mathcal{D}_k, h_k) \right]^{\frac{\alpha-1}{\alpha}} M^{\frac{1}{\alpha}} + \eta M$$

$$\le \mathsf{d}_\alpha(\mathcal{Q}) M^{\frac{1}{\alpha}} \epsilon^{\frac{\alpha-1}{\alpha}} + \eta M,$$

where for simplicity we write $\mathsf{d}_\alpha(\mathcal{Q}) = \max_{k \in [p]} \mathsf{d}_\alpha(\mathcal{D}_{k,\mathcal{Q}} \parallel \mathcal{D}_k)^{\frac{\alpha-1}{\alpha}}$. Applying this inequality to (13) yields

$$\mathcal{L}(\mathcal{D}_z, h_z^\eta) \le \mathsf{d}_\alpha(\mathcal{D}_z \parallel \mathcal{D}_{z,\mathcal{Q}})^{\frac{\alpha-1}{\alpha}} \mathcal{L}(\mathcal{D}_{z,\mathcal{Q}}, h_z^\eta)^{\frac{\alpha-1}{\alpha}} M^{\frac{1}{\alpha}}$$

$$\le \mathsf{d}_\alpha(\mathcal{D}_z \parallel \mathcal{D}_{z,\mathcal{Q}})^{\frac{\alpha-1}{\alpha}} \left[ \mathsf{d}_\alpha(\mathcal{Q}) M^{\frac{1}{\alpha}} \epsilon^{\frac{\alpha-1}{\alpha}} + \eta M \right]^{\frac{\alpha-1}{\alpha}} M^{\frac{1}{\alpha}}$$

$$\le \left[ \mathsf{d}_\alpha(\mathcal{D}_z \parallel \mathcal{D}_{z,\mathcal{Q}}) \, \mathsf{d}_\alpha(\mathcal{Q}) \right]^{\frac{\alpha-1}{\alpha}} M^{\frac{2\alpha-1}{\alpha^2}} \epsilon^{\frac{(\alpha-1)^2}{\alpha^2}}$$

$$+ \left[ \mathsf{d}_\alpha(\mathcal{D}_z \parallel \mathcal{D}_{z,\mathcal{Q}}) \right]^{\frac{\alpha-1}{\alpha}} M \eta^{\frac{\alpha-1}{\alpha}}.$$

Next, let $\mathcal{D}_\lambda$ be an arbitrary mixture of source domains, $\lambda \in \Delta$. Notice that Lemma 6 does not rely on any assumption of conditional probabilities, thus given fixed $\eta, \eta'$, we can still find $z$ such that $\mathcal{L}(\mathcal{D}_k, h_z^\eta) \le \mathcal{L}(\mathcal{D}_z, h_z^\eta) + \eta'$ for all $k \in [p]$, which implies that $\mathcal{L}(\mathcal{D}_\lambda, h_z^\eta) \le \mathcal{L}(\mathcal{D}_z, h_z^\eta) + \eta'$ for any $\lambda \in \Delta$. Thus, the choice of $z$ only depends on $\mathcal{D}_k, \forall k \in [p]$. This proves (12).

When the conditional probability distributions $\mathcal{D}_k(\cdot|x)$ do not depend on $k$, let $\mathcal{Q}(\cdot|x) = \mathcal{D}_k(\cdot|x)$, thus $\mathsf{d}_\alpha(\mathcal{D}_z \parallel \mathcal{D}_{z,\mathcal{Q}}) = 1$ and $\mathsf{d}_\alpha(\mathcal{Q}) = 1$. Setting $\alpha = +\infty$ and choosing $\eta, \eta'$ accordingly, we recover the result of Corollary 3.

Finally, by the proof of Corollary 12, for any $\lambda \in \Delta$,

$$\mathcal{L}(\mathcal{D}_T, h_z^\eta) \le \left[ \mathsf{d}_\alpha(\mathcal{D}_T \parallel \mathcal{D}_\lambda) \, \mathcal{L}(\mathcal{D}_\lambda, h_z^\eta) \right]^{\frac{\alpha-1}{\alpha}} M^{\frac{1}{\alpha}}$$

$$\le \left[ \mathsf{d}_\alpha(\mathcal{D}_T \parallel \mathcal{D}_\lambda) \, \mathcal{E}(\epsilon, \alpha, \eta, \eta') \right]^{\frac{\alpha-1}{\alpha}} M^{\frac{1}{\alpha}}.$$

Taking the infimum of the right-hand side over all $\mathcal{D}_\lambda \in \mathcal{D}$ completes the proof. $\square$

The main difference between Theorem 2 and Theorem 13 is the dependency of $z$: in Theorem 2, $z$ depends on a prefixed $\mathcal{Q}(\cdot|x)$, while in Theorem 13, $z$ only depends on $\mathcal{D}_k$. The guarantees in Theorem 13 ensure that we can first use $\mathcal{D}_k, k \in [p]$ to find a solution $z$ such that $h_z^\eta$ admits essentially the same loss on all source domains. Then, the performance of $h_z^\eta$ on arbitrary target distribution $\mathcal{D}_T$ relies on how close the source and target conditional probability distributions are to a *pivot* $\mathcal{Q}(\cdot|x)$, as well as on the divergence between $\mathcal{D}_T$ and $\mathcal{D}$.

## B.3  Probability model

In this section, we first present a series of general theoretical results for the probability model (P) in the same order as in Appendix B.1. Many of the them are similar to those for the regression model, except that we do not assume anything about the conditional probabilities throughout the proofs. In several instances, the proofs are syntactically the same as their counterparts in the regression model (R). In such cases, we do not reproduce them.

**Lemma 6.** *For any $\eta, \eta' > 0$, there exists $z \in \Delta$, with $z_k \neq 0$ for all $k \in [p]$, such that the following holds for the distribution-weighted combining rule $h_z^\eta$:*

$$\forall k \in [p], \quad \mathcal{L}(\mathcal{D}_k, h_z^\eta) \leq \sum_{j=1}^{p} z_j \mathcal{L}(\mathcal{D}_j, h_z^\eta) + \eta'. \tag{14}$$

*Proof.* The proof is syntactically the same as that for the regression model. $\qquad\square$

**Corollary 3.** *For any $\delta > 0$, there exist $\eta > 0$ and $z \in \Delta$, such that $\mathcal{L}(\mathcal{D}_\lambda, h_z^\eta) \leq \epsilon + \delta$ for any mixture parameter $\lambda \in \Delta$.*

*Proof.* Modifying the proof of Corollary 3 for the regression model gives

$$\mathcal{L}(\mathcal{D}_z, h_z^\eta) = \sum_{(x,y) \in \mathcal{X} \times \mathcal{Y}} \mathcal{D}_z(x,y) L(h_z^\eta(x,y))$$

$$= \sum_{(x,y)} \mathcal{D}_z(x,y) L\left( \sum_{k=1}^{p} \frac{z_k \mathcal{D}_k(x,y) + \eta \frac{\mathcal{U}(x,y)}{p}}{\mathcal{D}_z(x,y) + \eta \mathcal{U}(x,y)} h_k(x,y) \right).$$

By convexity of $L$, this implies that

$$\mathcal{L}(\mathcal{D}_z, h_z^\eta) \leq \sum_{(x,y)} \mathcal{D}_z(x,y) \sum_{k=1}^{p} \frac{z_k \mathcal{D}_k(x,y) + \eta \frac{\mathcal{U}(x,y)}{p}}{\mathcal{D}_z(x,y) + \eta \mathcal{U}(x,y)} L(h_k(x,y)).$$

Next, since $\frac{\mathcal{D}_z(x,y)}{\mathcal{D}_z(x,y) + \eta \mathcal{U}(x,y)} \leq 1$, the following holds:

$$\mathcal{L}(\mathcal{D}_z, h_z^\eta) \leq \sum_{(x,y)} \left( \sum_{k=1}^{p} \left( z_k \mathcal{D}_k(x,y) + \frac{\eta \mathcal{U}(x,y)}{p} \right) L(h_k(x,y)) \right)$$

$$= \sum_{k=1}^{p} z_k \mathcal{L}(\mathcal{D}_k, h_k) + \frac{\eta}{p} \sum_{k=1}^{p} \mathcal{L}(\mathcal{U}, h_k)$$

$$\leq \sum_{k=1}^{p} z_k \epsilon + \eta M = \epsilon + \eta M.$$

Now choose $z \in \Delta$ as in the statement of Lemma 4a. Then, the following holds for any mixture distribution $\mathcal{D}_\lambda$:

$$\mathcal{L}(\mathcal{D}_\lambda, h_z^\eta) = \sum_{k=1}^{p} \lambda_k \mathcal{L}(\mathcal{D}_k, h_z^\eta) \leq \sum_{k=1}^{p} \lambda_k (\mathcal{L}(\mathcal{D}_z, h_z^\eta) + \eta')$$

$$= \mathcal{L}(\mathcal{D}_z, h_z^\eta) + \eta' \leq \epsilon + \eta M + \eta'.$$

Setting $\eta = \frac{\delta}{2M}$ and $\eta' = \frac{\delta}{2}$ concludes the proof.

$\qquad\square$

Since we do not assume the conditional probabilities are the same across domains, we can directly prove the following theorem for the conditional probability model (P), which coincides with Theorem 1 when $\mathcal{D}_T \in \mathcal{D}$.

**Theorem 14.** *For any $\delta > 0$, there exist $\eta > 0$ and $z \in \Delta$, such that the following inequality holds for any $\alpha > 1$ and arbitrary target distribution $\mathcal{D}_T$:*

$$\mathcal{L}(\mathcal{D}_T, h_z^\eta) \leq \left[ (\epsilon + \delta) \, \mathsf{d}_\alpha(\mathcal{D}_T \parallel \mathcal{D}) \right]^{\frac{\alpha-1}{\alpha}} M^{\frac{1}{\alpha}} \qquad\qquad (P).$$

*Proof.* The proof is syntactically the same as that of Corollary 12 for the regression model. $\qquad\square$

**Corollary 15.** *Then, for any $\delta > 0$, there exist $\eta > 0$ and $z \in \Delta$, such that the following inequality holds for any $\alpha > 1$ and arbitrary target distribution $\mathcal{D}_T$:*

$$\mathcal{L}(\mathcal{D}_T, \widehat{h}_z^\eta) \leq \left[ (\widehat{\epsilon} + \delta) \, \mathsf{d}_\alpha(\mathcal{D}_T \parallel \widehat{\mathcal{D}}) \right]^{\frac{\alpha-1}{\alpha}} M^{\frac{1}{\alpha}},$$

*where $\widehat{\epsilon} = \max_{k \in [p]} \left[ \epsilon \, \mathsf{d}_\alpha(\widehat{\mathcal{D}}_k \parallel \mathcal{D}_k) \right]^{\frac{\alpha-1}{\alpha}} M^{\frac{1}{\alpha}}$, and $\widehat{\mathcal{D}} = \left\{ \sum_{k=1}^p \lambda_k \widehat{\mathcal{D}}_k : \lambda \in \Delta \right\}$.*

*Proof.* The proof is syntactically the same as that of Corollary 4 for the regression model. $\qquad\square$

## C  Specific theoretical analysis for the cross-entropy loss

Next, we give a specific theoretical analysis for the case of the cross-entropy loss. This is needed since the cross-entropy loss assumes normalized hypotheses. Thus, we are giving guarantees for the performance of normalized distribution-weighted predictor.

We will first assume that the conditional probability of the output labels is the same for all source domains, that is, for any $(x, y)$, $\mathcal{D}_k(y|x)$ is independent of $k$.

**Theorem 5.** *Assume that there exists $\mu > 0$ such that $\mathcal{D}_k(x, y) \geq \mu \mathcal{U}(x, y)$ for all $k \in [p]$ and $(x, y) \in \mathcal{X} \times \mathcal{Y}$. Then, for any $\delta > 0$, there exist $\eta > 0$ and $z \in \Delta$ such that $\mathcal{L}(\mathcal{D}_\lambda, \overline{h}_z^\eta) \leq \epsilon + \delta$ for any mixture parameter $\lambda \in \Delta$.*

*Proof.* By the proof of Corollary 3 for the probability model, for any mixture distribution $\mathcal{D}_\lambda$:

$$\mathcal{L}(\mathcal{D}_\lambda, h_z^\eta) \leq \epsilon + \eta M + \eta',$$

for some $\eta > 0, \eta' > 0$. For any $x \in \mathcal{X}$,

$$
\begin{aligned}
\sum_{y \in \mathcal{Y}} h_z^\eta(x, y) &= \sum_{y \in \mathcal{Y}} \sum_{k=1}^p \frac{z_k \mathcal{D}_k(x, y) + \frac{\eta \mathcal{U}(x,y)}{p}}{\mathcal{D}_z(x, y) + \eta \mathcal{U}(x, y)} h_k(x, y) \\
&\leq \sum_{y \in \mathcal{Y}} \sum_{k=1}^p \frac{z_k \mathcal{D}_k(x, y) + \frac{\eta \mathcal{U}(x,y)}{p}}{\mathcal{D}_z(x, y)} h_k(x, y) \\
&= 1 + \eta \left[ \frac{1}{p} \sum_{y \in \mathcal{Y}} \sum_{k=1}^p \frac{\mathcal{U}(x, y)}{\mathcal{D}_z(x, y)} h_k(x, y) \right].
\end{aligned}
\tag{15}
$$

By assumption, $\mathcal{D}_k(x, y) \geq \mu \mathcal{U}(x, y)$ for any $(x, y)$. Therefore $\mathcal{D}_z(x, y) \geq \mu \mathcal{U}(x, y)$ for any $z \in \Delta$. Since $0 \leq h_k(x, y) \leq 1$, equation (15) is further upper bounded by

$$\sum_{y \in \mathcal{Y}} h_z^\eta(x, y) \leq 1 + \eta \left[ \frac{1}{p} \sum_{y \in \mathcal{Y}} \sum_{k=1}^p \frac{\mathcal{U}(x, y)}{\mathcal{D}_z(x, y)} h_k(x, y) \right] \leq 1 + \frac{\eta |\mathcal{Y}|}{\mu}.$$

It follows that

$$
\begin{aligned}
\mathcal{L}(\mathcal{D}_\lambda, \overline{h}_z^\eta) &= \mathcal{L}(\mathcal{D}_\lambda, h_z^\eta) + \mathop{\mathbb{E}}_{x \sim \mathcal{D}_\lambda} \left[ \log \left( \sum_{y \in \mathcal{Y}} h_z^\eta(x, y) \right) \right] \leq \epsilon + \eta M + \eta' + \log \left( 1 + \frac{\eta |\mathcal{Y}|}{\mu} \right) \\
&\leq \epsilon + \eta \left( M + \frac{|\mathcal{Y}|}{\mu} \right) + \eta'.
\end{aligned}
$$

Setting $\eta = \frac{\delta}{2\left(M + \frac{|\mathcal{Y}|}{\mu}\right)}$ and $\eta' = \frac{\delta}{2}$ concludes the proof. $\qquad\square$

The analysis above depends on the key assumption that the conditional distributions $\mathcal{D}_k(y|x)$ are independent of $k$. When this assumption does not hold, we can show that there is a lower bound of $\log(p)$ on the generalization error $\mathcal{L}(\mathcal{D}_\lambda, \overline{h}_z^\eta)$. However, this lower bound coincides with that of convex combination rule (Lemma 10). In that case, one can use the following marginal distribution-weighted combination instead:

$$\widetilde{h}_z^\eta(x, y) = \sum_{k=1}^p \frac{z_k \mathcal{D}_k^1(x) + \eta \frac{\mathcal{U}^1(x)}{p}}{\sum_{j=1}^p z_j \mathcal{D}_j^1(x) + \eta \mathcal{U}^1(x)} h_k(x, y),
\tag{16}$$

where $\mathcal{D}_k^1(x)$ is the marginal distribution over $\mathcal{X}$, $\mathcal{D}_k^1(x) = \sum_{y \in \mathcal{Y}} \mathcal{D}_k(x, y)$, and $\mathcal{U}^1(x)$ is a uniform distribution over $\mathcal{X}$. Observe that $\widetilde{h}_z^\eta(x, y)$ is already normalized.

One can modify Theorem 2 to obtain generalization guarantees for $\widetilde{h}_z^\eta$ under distinct conditional probabilities assumption. Let $\mathcal{D}_T(x, y)$, $\epsilon_\alpha(\mathcal{Q})$ and $\mathcal{D}_{P,Q}$ be defined as before.

**Theorem 16.** *For any $\delta > 0$, there exist $\eta > 0$ and $z \in \Delta$ such that the following inequality holds for any $\alpha, \beta > 1$ and arbitrary target distribution $\mathcal{D}_T$:*

$$\mathcal{L}(\mathcal{D}_T, \widetilde{h}_z^\eta) \leq \left[ \left( \epsilon_\alpha(\mathcal{Q}) + \delta \right) \mathsf{d}_\beta(\mathcal{D}_T \parallel \mathcal{D}_{P,\mathcal{Q}}) \right]^{\frac{\beta-1}{\beta}} M^{\frac{1}{\beta}}.$$

*Proof.* The proof is syntactically the same as that of Theorem 2. $\qquad\qquad\qquad\square$

Finally, we can extend Theorem 5 and Theorem 16 to the case where only estimate distributions $\widehat{\mathcal{D}}_k$s are available, and the predictor $\overline{\widetilde{h}_z^\eta}$ and $\widetilde{\overline{h}_z^\eta}$ based on the estimates $\widehat{\mathcal{D}}_k$ still admit favorable guarantees. The results and proofs are similar to proving Corollary 4 from Corollary 12 in the regression model, thus omitted here.

# D DC-decomposition

In this section we give the full proofs for the DC-decompositions presented in Section 4.2.

## D.1 Regression model

**Proposition 7.** *Let $L$ be the squared loss. Then, for any $k \in [p]$, $\mathcal{L}(\mathcal{D}_k, h_z^\eta) - \mathcal{L}(\mathcal{D}_z, h_z^\eta) = u_k(z) - v_k(z)$, where $u_k$ and $v_k$ are convex functions defined for all $z$ by*

$$u_k(z) = \mathcal{L}\left(\mathcal{D}_k + \eta \mathcal{U}^1 \mathcal{D}_k(\cdot|x), h_z^\eta\right) - 2M \sum_x (\mathcal{D}_k^1 + \eta \mathcal{U}^1)(x) \log K_z(x),$$

$$v_k(z) = \mathcal{L}\left(\mathcal{D}_z + \eta \mathcal{U}^1 \mathcal{D}_k(\cdot|x), h_z^\eta\right) - 2M \sum_x (\mathcal{D}_k^1 + \eta \mathcal{U}^1)(x) \log K_z(x).$$

*Proof.* First, observe that $(h_z^\eta(x) - y)^2 = f_z(x, y) - g_z(x)$, where for every $(x, y) \in \mathcal{X} \times \mathcal{Y}$, $f_z$ and $g_z$ are convex functions defined for all $z$:

$$f_z(x, y) = (h_z^\eta(x) - y)^2 - 2M \log K_z(x),$$
$$g_z(x) = -2M \log K_z(x).$$

This is true because the Hessian matrix of $f_z$ and $g_z$ are

$$H_{f_z} = \frac{2}{K_z^2}\left[h_{D,z}h_{D,z}^T + \left(M - (y - h_z^\eta)^2\right)DD^T\right],$$

$$H_{g_z} = \frac{2M}{K_z^2}DD^T,$$

where $h_{D,z}$ is a $p$-dimensional vector defined as $[h_{D,z}]_k = \mathcal{D}_k(h_k + y - 2h_z^\eta)$ for $k \in [p]$, and $D = (\mathcal{D}_1, \mathcal{D}_2, \ldots, \mathcal{D}_p)^T$. Using the fact that $M \geq (y - h_z^\eta)^2$, $H_{f_z}$ and $H_{g_z}$ are positive semidefinite matrices, therefore $f_z, g_z$ are convex functions of $z$.

Thus, $u_k(z) = \sum_{(x,y)}(\mathcal{D}_k^1 + \eta \mathcal{U}^1)(x)\mathcal{D}_k(y|x)f_z(x, y)$ is convex. Similarly, we can write the second term of $v_k(z)$ as $\sum_x (\mathcal{D}_k^1 + \eta \mathcal{U}^1)(x)g_z(x)$, it is convex. Using the notation previously defined, we can write the first term of $v_k(z)$ as

$$\mathcal{L}(\mathcal{D}_z + \eta \mathcal{U}^1 \mathcal{D}_k(\cdot|x), h_z^\eta) = \sum_x \frac{J_z(x)^2}{K_z(x)} - 2\,\mathbb{E}(y|x)J_z(x) + \mathbb{E}(y^2|x)K_z(x).$$

The Hessian matrix of $J_z^2/K_z$ is

$$\nabla_z^2\left(\frac{J_z^2}{K_z}\right) = \frac{1}{K_z}(h_D - h_z^\eta D)(h_D - h_z^\eta D)^T$$

where $h_D = (h_1\mathcal{D}_1, h_2\mathcal{D}_2, \ldots, h_p\mathcal{D}_p)^T$ and $D = (\mathcal{D}_1, \mathcal{D}_2, \ldots, \mathcal{D}_p)^T$. Thus $J_z^2/K_z$ is convex. $-2\,\mathbb{E}(y|x)J_z(x) + \mathbb{E}(y^2|x)K_z(x)$ is an affine function of $z$ and is therefore convex. Therefore the first term of $v_k(z)$ is convex, which completes the proof. $\square$

## D.2 Probability model

**Proposition 8.** *Let $L$ be the cross-entropy loss. Then, for $k \in [p]$, $\mathcal{L}(\mathcal{D}_k, h_z^\eta) - \mathcal{L}(\mathcal{D}_z, h_z^\eta) = u_k(z) - v_k(z)$, where $u_k$ and $v_k$ are convex functions defined for all $z$ by*

$$u_k(z) = -\sum_{x,y}\left[\mathcal{D}_k(x, y) + \eta \mathcal{U}(x, y)\right]\log J_z(x, y),$$

$$v_k(z) = \sum_{x,y} K_z(x, y)\log\left[\frac{K_z(x, y)}{J_z(x, y)}\right]$$
$$- \left[\mathcal{D}_k(x, y) + \eta \mathcal{U}(x, y)\right]\log K_z(x, y).$$

*Proof.* Using the notation previously introduced, we can now write

$$
\begin{aligned}
&\mathcal{L}(\mathcal{D}_k, h_z^\eta) - \mathcal{L}(\mathcal{D}_z, h_z^\eta) \\
&= \operatorname*{\mathbb{E}}_{(x,y)\sim\mathcal{D}_k}\big[-\log h_z^\eta(x,y)\big] - \operatorname*{\mathbb{E}}_{(x,y)\sim\mathcal{D}_z}\big[-\log h_z^\eta(x,y)\big] \\
&= \sum_{x,y}\big(\mathcal{D}_z(x,y) - \mathcal{D}_k(x,y)\big)\log\left[\frac{J_z(x,y)}{K_z(x,y)}\right] \\
&= \sum_{x,y}\big[K_z(x,y) - (\mathcal{D}_k(x,y) + \eta\,\mathcal{U}(x,y))\big]\log\left[\frac{J_z(x,y)}{K_z(x,y)}\right] \\
&= u_k(z) - v_k(z).
\end{aligned}
$$

$u_k$ is convex since $-\log J_z$ is convex as the composition of the convex function $-\log$ with an affine function. Similarly, $-\log K_z$ is convex, which shows that the second term in the expression of $v_k$ is a convex function. The first term can be written in terms of the unnormalized relative entropy:[1]

$$
\begin{aligned}
&\sum_{x,y} K_z(x,y)\log\left[\frac{K_z(x,y)}{J_z(x,y)}\right] \\
&= B(K_z \parallel J_z) + \sum_{(x,y)}(K_z - J_z)(x,y).
\end{aligned}
$$

The unnormalized relative entropy $B(\cdot \parallel \cdot)$ is jointly convex (Cover and Thomas, 2006),[2] thus $B(K_z \parallel J_z)$ is convex as the composition of the unnormalized relative entropy with affine functions (for each of its two arguments). $(K_z - J_z)$ is an affine function of $z$ and is therefore convex too. $\square$

Figure 2: Synthetic global loss versus iteration for squared loss. Our solution converges to the global optimum of zero.

| (a) | (b) |

Figure 3: (a) Artificial dataset for cross-entropy loss, with three domains (red, green and blue) and three categories (triangle, square, circle). (b) Artificial dataset global loss versus iteration for cross-entropy loss. We empirically find that our solution converges to the global optimum of zero.

## E    Additional experiment results

In this section we provide experiment results on artificial datasets to show that our global objective indeed approaches the known optimal of zero with DC-programming algorithm, for both squared loss and cross-entropy loss. We also provide details of our density estimation procedure on the real-world applications, as well as additional experiment results to show that our distribution-weighted predictor DW is robust across various test data mixtures.

### E.1    Artificial dataset

We first evaluated our algorithm on synthetic datasets, for both squared loss and cross-entropy loss.

Consider the following multiple source domain study by Mansour et al. (2009a). Let $g_1, g_2, g_3, g_4$ denote the Gaussian distributions with means $(1,1)$, $(-1,1)$, $(-1,-1)$, and $(1,-1)$ and unit variance respectively. Each domain was generated as a uniform mixture of Gaussians: $\mathcal{D}_1$ from $\{g_1, g_2, g_3\}$ and $\mathcal{D}_2$ from $\{g_2, g_3, g_4\}$. The labeling function is $f(x_1, x_2) = x_1^2 + x_2^2$. We trained linear regressors for each domain to produce base hypotheses $h_1$ and $h_2$. Finally, as the true distribution is known for this artificial example, we directly use the Gaussian mixture density function to generate our $\mathcal{D}_k$s.

With this data source, we used our DC-programming solution to find the optimal mixing weights $z$. Figure 2 shows the global objective value (of Problem 4) vs number of iterations with the uniform initialization $z_0 = [1/2, 1/2]$. Here, the overall objective approaches $0.0$, the known global minimum. To verify the robustness of the solution, we have experimented with various initial conditions and found that the solution converges to the global solution in each case.

We next evaluate our algorithm on cross-entropy loss. Here we generate the two-dimensional dataset shown in Figure 3a, which has three domains, denoted in the colors red, green, and blue, and three

Table 5: MSE on sentiment analysis dataset: target domain as various combinations of two domains.

| | Test Data | | | | | |
|---|---|---|---|---|---|---|
| | KD | BE | KB | KE | DB | DE |
| K | 1.83±0.08 | 1.99±0.10 | 1.87±0.08 | 1.57±0.06 | 2.25±0.08 | 1.94±0.10 |
| D | 1.95±0.07 | 2.11±0.07 | 2.12±0.07 | 2.11±0.05 | 1.95±0.06 | 1.94±0.06 |
| B | 2.10±0.09 | 1.99±0.08 | 1.96±0.07 | 2.21±0.06 | 1.87±0.07 | 2.13±0.05 |
| E | 2.00±0.09 | 1.95±0.07 | 2.05±0.05 | 1.60±0.05 | 2.36±0.07 | 1.91±0.07 |
| unif | 1.73±0.06 | 1.74±0.07 | 1.74±0.05 | 1.62±0.04 | 1.85±0.05 | 1.73±0.06 |
| KMM | 1.83±0.07 | 1.82±0.07 | 1.78±0.12 | 1.65±0.10 | 1.97±0.13 | 1.88±0.08 |
| DW | **1.62±0.07** | **1.61±0.08** | **1.59±0.05** | **1.47±0.04** | **1.75±0.05** | **1.64±0.05** |

Table 6: MSE on the sentiment analysis dataset: target domain as various mixture of four domains: $(\mathbf{0.4}, 0.2, 0.2, 0.2)$, $(0.2, \mathbf{0.4}, 0.2, 0.2)$, $(0.2, 0.2, \mathbf{0.4}, 0.2)$, $(0.2, 0.2, 0.2, \mathbf{0.4})$ of K, D, B, E respectively.

| | Test Data | | | |
|---|---|---|---|---|
| | KDBE | KDBE | KDBE | KDBE |
| K | 1.78±0.05 | 1.94±0.10 | 1.96±0.08 | 1.84±0.07 |
| D | 2.02±0.10 | 1.98±0.10 | 2.06±0.11 | 2.05±0.09 |
| B | 2.01±0.12 | 2.01±0.14 | 1.94±0.14 | 2.06±0.11 |
| E | 1.93±0.08 | 2.04±0.10 | 2.08±0.10 | 1.89±0.08 |
| unif | 1.69±0.06 | 1.74±0.07 | 1.75±0.08 | 1.70±0.06 |
| KMM | 1.83±0.12 | 1.92±0.14 | 1.87±0.15 | 1.85±0.13 |
| DW | **1.55±0.08** | **1.62±0.08** | **1.59±0.09** | **1.56±0.08** |

categories, denoted as squares, circles, and triangles. Each domain is generated according to a Gaussian mixture model, one mixture per category, with random means. The means of each corresponding category across domains are related according to a random fixed orthonormal transformation. Finally, the covariance of each mixture is diagonal and fixed across categories. We choose covariance magnitudes of 0.05, 0.05, and 0.3 for the red, green, and blue domains, respectively. We then train a logistic regression classifier per domain to produce score functions, $h_k$. Finally, as the true distribution is known for this artificial example, we forgo density estimation and use the Gaussian mixture density function to generate our $\mathcal{D}_k$s.

With this data source, we use our DC-programming solution to find the optimal mixing weights, $z$. Since only each convex sub-problem is guaranteed to converge, Figure 3b reports this global loss vs iteration when initializing $z_0 = 1/p$, uniform weights. Here, the overall objective approaches 0.0, the known global minimum. To verify the robustness of the solution, we have experimented with various initial conditions and found the solution converges to the global solution from each case.

### E.2 Sentiment analysis task for squared loss

We begin by detailing our density estimation method for the sentiment analysis experiment. We first used the same vocabulary defined for feature extraction to train a separate bigram statistical language model for each domain, using the OpenGrm library (Roark et al., 2012). Next, we randomly draw a sample set $S_k$ of 10,000 sentences from each bigram language model. We define $\widehat{\mathcal{D}}_k$ to be the empirical distribution of $S_k$, which is a very close estimate of marginal distribution of the language model, thus it is also a good estimate of $\mathcal{D}_k$. We approximate the label of a randomly generated sample $x_i$ by taking the average of the $h_k$s: $y_i = \sum_{\{k:x_i \in S_k\}} h_k(x_i)/|\{k: x_i \in S_k\}|$. These randomly drawn samples were used to find the fixed-point $z$.

Note that we only use estimates of the marginal distributions (language models) to find $z$ and do not use any labels. We use the original product review text and rating labels for testing. Their densities $\widehat{\mathcal{D}}_k$ were estimated by the bigram language models directly, therefore a close estimate of $\mathcal{D}_k$.

Next we compare DW to accessible predictors on various test mixture domains. Table 5 shows MSE on all combinations of two domains. Table 6, 7 reports MSE on additional test mixture domains. The first four target mixtures correspond to various orderings of $(0.4, 0.2, 0.2, 0.2)$. The next six target mixtures correspond to various orderings of $(0.3, 0.3, 0.2, 0.2)$. In column titles we bold the domain(s) with highest weight.

Table 7: MSE on the sentiment analysis dataset: target domain as various mixture of four domains: $(\mathbf{0.3}, \mathbf{0.3}, 0.2, 0.2)$, $(\mathbf{0.3}, 0.2, \mathbf{0.3}, 0.2)$, $(\mathbf{0.3}, 0.2, 0.2, \mathbf{0.3})$, $(0.2, \mathbf{0.3}, \mathbf{0.3}, 0.2)$, $(0.2, \mathbf{0.3}, 0.2, \mathbf{0.3})$, $(0.2, 0.2, \mathbf{0.3}, \mathbf{0.3})$ of `K, D, B, E` respectively.

| | Test Data | | | | | |
|---|---|---|---|---|---|---|
| | KDBE | KDBE | KDBE | KDBE | KDBE | KDBE |
| K | 1.86±0.10 | 1.87±0.07 | 1.79±0.08 | 1.96±0.10 | 1.89±0.10 | 1.89±0.08 |
| D | 2.01±0.13 | 2.05±0.12 | 2.04±0.12 | 2.03±0.12 | 2.02±0.13 | 2.06±0.12 |
| B | 2.01±0.15 | 1.98±0.14 | 2.05±0.13 | 1.98±0.15 | 2.04±0.14 | 2.01±0.13 |
| E | 2.00±0.10 | 2.01±0.09 | 1.91±0.08 | 2.08±0.10 | 1.97±0.08 | 1.99±0.08 |
| unif | 1.72±0.09 | 1.72±0.08 | 1.69±0.07 | 1.75±0.08 | 1.72±0.08 | 1.73±0.08 |
| KMM | 1.85±0.16 | 1.86±0.14 | 1.85±0.15 | 1.90±0.14 | 1.89±0.16 | 1.90±0.14 |
| DW | **1.58±0.10** | **1.57±0.10** | **1.55±0.09** | **1.61±0.10** | **1.59±0.08** | **1.58±0.09** |

In all these experiments, our distribution-weighted predictor `DW` outperforms all competing baselines: the source only baselines for each domain, `K, D, B, E`, a uniform weighted predictor `unif`, and `KMM`.

### E.3 Recognition tasks for cross-entropy loss

Here, we describe our density estimation technique for the object recognition task.

To estimate the per domain densities, we first extract per image features using the in-domain ConvNet model, and then estimate the marginal distribution $\mathcal{D}_k^1(x)$ over the per domain collection of features, using non-parametric kernel density estimation with a Gaussian kernel and a cross-validated bandwidth parameter. We use estimated marginals $\widehat{\mathcal{D}^1}_k$ instead of estimated joint distributions $\widehat{\mathcal{D}}_k$, because when the conditional probabilities are the same across domains and when $\eta \to 0$, $h_z^\eta(x, y)$ converges to a normalized predictor $\widetilde{h}_z(x, y) = \sum_{k=1}^p \frac{z_k \mathcal{D}_k^1(x)}{\sum_{j=1}^p z_j \mathcal{D}_j^1(x)} h_k(x, y)$. Thus in our experiments, we approximate $\widehat{h}_z^\eta(x, y)$ with $\widetilde{h}(x, y)$ using our estimated marginal distributions $\widehat{\mathcal{D}^1}_k(x)$.

# F  Rényi Divergence

The Rényi Divergence measures the divergence between two distributions. It is parameterized by $\alpha \in [0, +\infty]$ and denoted by $D_\alpha$. The $\alpha$-Rényi Divergence of two distributions $\mathcal{D}$ and $\mathcal{D}'$ is defined by

$$D_\alpha(\mathcal{D} \parallel \mathcal{D}') = \frac{1}{\alpha - 1} \log \sum_{(x,y) \in \mathcal{X} \times \mathcal{Y}} \mathcal{D}(x,y) \left[ \frac{\mathcal{D}(x,y)}{\mathcal{D}'(x,y)} \right]^{\alpha - 1}, \tag{17}$$

where, for $\alpha \in \{0, 1, +\infty\}$, the expression is defined by taking the limit. It can be shown that the Rényi Divergence is always non-negative and that for any $\alpha > 0$, $D_\alpha(\mathcal{D} \parallel \mathcal{D}') = 0$ iff $\mathcal{D} = \mathcal{D}'$ (Arndt, 2004). We will denote by $d_\alpha(\mathcal{D} \parallel \mathcal{D}')$ the exponential:

$$d_\alpha(\mathcal{D} \parallel \mathcal{D}') = e^{D_\alpha(\mathcal{D} \parallel \mathcal{D}')} = \left[ \sum_{(x,y) \in \mathcal{X} \times \mathcal{Y}} \frac{\mathcal{D}^\alpha(x,y)}{\mathcal{D}'^{\alpha - 1}(x,y)} \right]^{\frac{1}{\alpha - 1}}. \tag{18}$$

The Rényi divergence (and $d_\alpha(\mathcal{D} \parallel \mathcal{D}')$) is a non-decreasing function of $\alpha$; in particular, the following inequality holds:

$$d_\alpha(\mathcal{D} \parallel \mathcal{D}') \leq d_\infty(\mathcal{D} \parallel \mathcal{D}') = \sup_{(x,y) \in \mathcal{X} \times \mathcal{Y}} \left[ \frac{\mathcal{D}(x,y)}{\mathcal{D}'(x,y)} \right]. \tag{19}$$

# G Related work on multiple source adaptation (MSA)

We give an extensive discussion of related work on multiple source adaptation (MSA) problem here, and point out how our scenario and our results are distinct from most previous works.

The learning scenario that we consider is distinct from and is more challenging than the one considered by many other existing multiple source adaptation (MSA) studies:

1. We assume that the learner does not have access to and therefore cannot *combine all the source labeled data* together to jointly train a target predictor. This is a very realistic assumption with legitimate reasons, such as data privacy, storage limitation, etc. Instead, the learner is only given pre-trained models, density estimations from the source domains, and a small subset of combined source labeled data.

2. We are not given any target data, even unlabeled, and we are competing against any target mixture distribution. This is a significantly more difficult problem. Remarkably, the learner only needs to run our algorithm DW once to obtain a single predictor, which is guaranteed to perform well on *any* target mixture. This is also verified experimentally.

To our knowledge, there is no discussion of this learning scenario other than by Mansour et al. (2008, 2009a). On the contrary, most MSA algorithms require access to the full combined source data to jointly train a predictor. In addition, many MSA algorithms require a set of labeled or unlabeled data from the target domain, and the solution only performs well on that specific target domain. If the target domain changes, the learner has to rerun the algorithm to find a new solution. Besides, many MSA algorithms do not admit theoretical guarantees, while we provide a careful analysis and series of strong theoretical guarantees for our algorithm DW.

In what follows, we categorize and discuss previous works on MSA problem by their learning scenarios.

**Combine source data.** Khosla et al. (2012) considered a similar setting where the learner trains a single predictor for any target domain and where the learner has access to source data but not target data. However, Khosla et al. (2012) combine all the source data to jointly train the final predictor and a large set of combined data is necessary for a good predictor. Additionally, the solution of Khosla et al. (2012) only works for linear functions, a very limited family of hypotheses. DW does not combine all the source data, and works for hypotheses of any form. Blanchard et al. (2011) presented MSA algorithms with theoretical guarantees. However, it combines all source data and target data to learn a final predictor. This paper also makes the strong assumption that the source and the target domains are i.i.d. realizations of some distribution, and their learning guarantee is with respect to that distribution. DW makes no assumption about the relationship between the source domains. Hoffman et al. (2012) considered multiclass classification problem where the predicted label for a novel test point is determined by a weighted sum of probabilities of each category given that the test point comes from a particular source domain. The weights are the predicted probability that the test point belongs to each source domain, which are learned via SVM on all source data combined. Zhang et al. (2015) considered a causal view of MSA where label $Y$ is the cause for features $X$, and learned a weighted combination of source conditional probabilities ($\mathbb{P}_{X|Y}$) by minimizing the maximum mean discrepancy (MMD) on the combined source data. Muandet et al. (2013) proposed Domain-Invariant Component Analysis (DICA) to transform features onto a low dimensional subspace by minimizing dissimilarity across multiple source domains, while preserving relationship between features and label. The projection is learned via a kernel-based optimization on all source data combined. Recently, Pei et al. (2018) extended domain adversarial learning techniques for the multiple source setting.

**Use labeled target data.** Duan et al. (2009, 2012) considered a somewhat similar problem where the learner leverages pre-trained predictors from the source domains to learn a good predictor on the target domain. However, they assume plenty of unlabeled target data to form a meaningful regularizer and they also assume a small set of labeled target data. Their solution directly depends on the labeled and unlabeled target data and is of course only useful for that specific target. Yang et al. (2007) also considered the problem of combining pre-trained classifiers from multiple auxiliary datasets to adapt to a target dataset, and the solution is to learn a good linear combinations of auxiliary classifiers using Adaptive SVMs on the labeled target data.

**Others.** Crammer et al. (2008) dealt with a problem with multiple sources distinct from domain adaptation where the sources have the same input distribution but can have different labels, modulo some disparity constraints. (See also discussion by Mansour et al. (2009a)). Gong et al. (2012) proposed a Rank of Domain (ROD) metric that ranks multiple source domains by how likely they are to adapt well to a target domain. Gong et al. (2013a) learned domain-invariant features by first constructing multiple auxiliary tasks based on landmarks within the source data, and then learning new feature representations from each auxiliary task. Gong et al. (2013b) proposed to discover multiple latent domains by maximizing two key properties, distinctiveness and learnability, between latent domains. Xu et al. (2014) also considered the problem of discovering latent domains, and proposed to learn exemplar-SVMs with low-rank structure.