[Reviews · NeurIPS 2018]

Reviewer 1



In this paper, a multi-source domain adaptation algorithm is proposed, which is derived from the theory by Mansour et al. 2008 that to learn a weighted combination of source distributions. The authors discussed the proposed algorithm in the stochastic scenario, which is important for being applied into deep neural networks. Experiments on a few benchmarks show that the proposed approach outperforms a few baselines. Generally, the paper is well written, and the algorithm development is reasonable. However, multiple source domain adaptation is not a new problem, and many approaches have been proposed to tackle with this task based on various principles. Unfortunately, the authors did not present a thorough discussion the connection with existing related works, [Hoffman et al. 2012, Gong et al 2013b, Xu et al. 2014, Zhang et al. 2015] and reference below. No experimental comparison is provided either. It is interesting to see an instantiation of Mansour’s theory, but the work in its current form is not sufficiently convincing. [ref-1] Lixin Duan, Dong Xu, Ivor Wai-Hung Tsang. Domain Adaptation from Multiple Sources: A Domain-Dependent Regularization Approach. In IEEE Transactions on Neural Networks and Learning Systems (T-NNLS), 2012. [ref-2] Aditya Khosla, Tinghui Zhou, Tomasz Malisiewicz, Alexei A. Efros, Antonio Torralba. Undoing the Damage of Dataset Bias. In ECCV 2012. [ref-3] Saeid Motiian, Marco Piccirilli, Donald A. Adjeroh, Gianfranco Doretto. Unified Deep Supervised Domain Adaptation and Generalization. In ICCV 2017.

Reviewer 2



Summary of the paper: This paper is interested in the problem of multiple-source adaptation. More precisely it is interested in the setting where a good model and an estimation of the distribution is available for several source domains. The goal is then to learn a good combination of the models to accurately classify target examples drawn from a mixture of the source domains. On the one hand the paper extends the theoretical results of [Mansour et al., 2008] to the stochastic setting where there exists a distribution over the joint input-output space. On the other hand it proposes an algorithm to correctly combine the models available from the different domains. The performance of the approach is empirically demonstrated on several tasks. Main comments: Pros: - The paper is very well written and easy to read - The setting considered is more general than existing works - The proposed algorithm to estimate the target mixture is new, theoretically sound (Lemma 4) and works well in practice Cons: - The theoretical results are a bit incremental [Mansour et al. 2008] - The practical usefulness of the method is not so clear as it requires an estimate of the distribution and a model for each domain Detailed comments: This paper is well written and, despite being quite technical, is easy to follow. In the experiment displayed in Figure 1 and introduced Lines 246 to 251, I would have appreciated to see the results of the predictor used in Mansour et al. [2008]. Even it is not a real solution, it remains a good baseline to assess the limit of the method considered, that is it is probably one of the best achievable results using distribution-weighted combination of the source models. The theoretical results are largely based on the previous work of Mansour et al. [2008] and mainly consist of an extension from the deterministic to the stochastic setting. In particular it considers a more realistic setting where the conditional probabilities of each output given an input might be different for the source and the target domain. One of the main drawbacks of the proposed approach is that it assumes that the underlying distribution of the inputs is available for each domain which might not be very realistic in practice. This issue can be overcome by using a large number of unlabelled data from each domain to estimate the underlying distribution. In which kind of applications would this kind of setting (a good model and a set of unlabelled examples for each domain) be reasonable? Some other minor comments: - In the main paper, Line 111: I would have appreciated to have a bit more intuition about the meaning of \epsilon_T and not a simple definition, that is what does it represent or account for? - In the supplementary, in the inequality following Line 440: It should be \hat{h} rather than h. After the rebuttal: The rebuttal solved my concerns regarding the practical usefulness of the method. However I still think that the contribution is a bit incremental and I agree with Reviewer 1 that a more extensive empirical evaluation would have been beneficial.

Reviewer 3



This paper proposes a number of theoretical findings for the multiple source adaptation problems. Specifically, it presents new methods to determine the distribution-weighted combination solution for the cross-entropy loss and other similar losses. Extensive experimental studies have been conducted to verify these findings. Overall, this paper is well-written. The theoretical analyses seem to be correct, although I cannot fully check all the proofs. The experimental studies are extensive and well support the proposed theoretical findings. My only concern on this paper is the lack of discussions on the related works. The current paper focuses on the discussion of gaps with Mansour et al. [2008,2009], and it further extends to deal with these gaps. However, there are many other existing works studying the theoretical analyses on the multiple source domains, e.g. [ref1], [ref2], and [ref3]. What are the differences of the current paper with these existing works? Some of gaps in Mansour et al. [2008,2009] may have already been considered in other works, for instance, [ref3] also considers the joint distribution of the input and output. The authors should pay more attention to the discussion on those related works, to further motivate the paper and highlight the significance of the paper. [ref1] Learning from Multiple Sources [ref2] Domain Adaptation from Multiple Sources via Auxiliary Classifiers [ref3] Generalizing from Several Related Classification Tasks to a New Unlabeled Sample ####### After reading the authors' response, I am satisfied with the extended discussions on the related works. However, after discussions with other reviewers, I also agree with reviewer 1 on the point that more comprehensive empirical studies should be done.